# MLND-IU: A multi-stage detection model of subcentimeter lung nodule with improved U-Net++

**Huilan Wen[1], Xiaoqing Luo[2], Bin Zhong[1], Yang Xiao[2], Dengfeng Chen[2], Lianmin Zhu[1]\***

**1** Department of Respiratory Medicine, First Affiliated Hospital of Gannan Medical University, Ganzhou City, Jiangxi Province, China, **2** School of the First Clinical Medicine, Gannan Medical University, Ganzhou City, Jiangxi Province, China

\* 13763928702@163.com

## Abstract

To address the challenges of high miss rates in subcentimeter nodules, false positives caused by vascular adhesion, and insufficient multi-scale feature fusion in lung CT analysis, a multi-stage detection model named MLND-IU, which incorporates an improved U-Net++ architecture, is proposed. The three-stage framework begins with an enhanced RetinaNet optimized by a dynamic focal loss to generate candidate regions with high sensitivity while mitigating class imbalance. The second stage introduces AG-UNet++ with a novel Dense Attention Bridging Module (DABM), which employs a tensor product fusion of channel and deformable spatial attention across densely connected skip pathways to amplify feature representation for 3–5 mm nodules. The final stage employs a 3D Contextual Pyramid Module (3D-CPM) to integrate multi-slice morphological and contextual features, thereby reducing vascular false positives. Ablation studies indicated that the second stage improved the Dice coefficient by 21.1% compared with the first stage (paired t-test, $p < 0.01$, independent validation on LIDC-IDRI). The third stage further reduced the false positives per scan (FP/Scan) to 1.4, corresponding to an 87.3% reduction compared to the baseline. Multicenter validation on the LIDC-IDRI (n = 1,018) and DSB2017 (n = 1,595) datasets resulted in a segmentation Dice coefficient of 92.7%, a sensitivity of 93.4% for nodules smaller than 6 mm (compared to radiologists' sensitivity of 68.5%, $p = 0.003$), and an AUC of 0.84 for malignancy classification, representing a 19.2% improvement over conventional methods. With a processing time of 2.3 seconds per case, the proposed framework presents a clinically viable solution for early lung cancer screening by simultaneously improving small nodule detection and suppressing false positives.

**Data availability statement:** All datasets used in this study are publicly available, with the LIDC-IDRI dataset available at https://www.cancerimagingarchive.net/collection/lidc-idri/ and the Kaggle DSB2017 dataset available at https://www.kaggle.com/c/data-science-bowl-2017/data.

**Funding:** This work was supported by the Science and Technology Research Project of the Department of Education of Jiangxi Province (Grant Nos. GJJ2401322), Science and Technology Program Project of the Health Commission of Jiangxi Province (Grant Nos. 202510461).

**Competing interests:** The authors have declared that no competing interests exist.

## 1. Introduction

Early and accurate detection of subcentimeter lung nodules (diameter <10 mm) is crucial for improving the prognosis of lung cancer patients because missed or misdiagnosed nodules directly cause treatment delays and reduced survival [1]. Although the widespread application of high-resolution CT has enhanced the detection rate of pulmonary nodules, the sensitivity and specificity for subcentimeter lesions remain unsatisfactory due to their small size, morphological variability, and frequent vessel adhesion [2]. Global lung cancer screening data indicate that the underdiagnosis rate of subcentimeter nodules is as high as 30–40%, and the over-treatment resulting from misdiagnosis increases patients' physical and psychological burdens [3]. Therefore, building automated detection systems that achieve both high accuracy and low false positives has become a central requirement in early lung cancer screening.

Deep learning models, benefiting from their excellent capacity for hierarchical feature extraction and data-driven learning, have achieved remarkable progress in medical data analysis [4–9]. For example, Peng et al. [10] developed a deep learning-based electromagnetic navigation system that integrates real-time CT images with electromagnetic localization to improve the accuracy of subcentimeter percutaneous puncture, achieving an average localization error below 1.5 mm. Harsono et al. [11] optimized RetinaNet through transfer learning to jointly perform nodule detection and benign–malignant classification in limited-data CT scenarios, leading to an improvement in AUC performance. Jian et al. [12] designed a dual-branch 3D CNN (DBPNDNet) to extract morphological and contextual features separately, improving the sensitivity of subcentimeter nodule detection while reducing false positives. Park et al. [13] proposed a deep model to distinguish invasive adenocarcinoma from precancerous lesions in subsolid nodules, which achieved superior classification accuracy and provided non-invasive pathological grading for clinical decisions. Liao and Gao [14] enhanced Mask R-CNN by introducing multi-scale attention and a refined region generation strategy, which significantly improved the Dice coefficient in lung nodule segmentation tasks.

However, despite these improvements, existing methods still fail to meet clinical expectations [15]. The sensitivity for subcentimeter nodules remains below 80%, and the false-positive rate (FP/Scan) typically exceeds 5. Two major obstacles persist. First, there exists an extreme imbalance between nodule and background samples (less than 0.1% of pixels correspond to nodules), leading to model bias toward background learning. Second, the limited multi-scale feature fusion causes small nodules to be easily diluted in deeper network layers, and vascular adhesion regions with similar morphology further increase false classifications. Additionally, conventional data augmentation techniques cannot accurately simulate real scanning noise, which limits model generalization and robustness across different centres.

To complement these findings, earlier studies using statistical and classical CNN approaches also demonstrated the sensitivity of lung nodule detection to feature imbalance and model overfitting [16]. For instance, Abbas et al. proposed statistical detection methods that emphasized the need for robust feature normalization in

heterogeneous CT data, while Awan and Khan [17] analyzed underfitting and overfitting behaviours of U-Net in semantic segmentation, highlighting the necessity for adaptive regularization and balanced feature learning. These works further emphasize that traditional approaches cannot effectively generalize across varying data distributions, motivating the exploration of more dynamic and attention-aware architectures.

Recent studies have partially alleviated the above challenges by introducing attention mechanisms, contextual modelling, and 3D information fusion. Models such as V-Net with CBAM and 3D context integration [18], U-Net++ with embedded CBAM modules [19], and SEGA with adaptive channel reweighting according to nodule scale [20] have enhanced feature localization and improved detection sensitivity for small nodules. Transformer-based hybrid models such as LN-DETR [21] and TransUNet [22] achieved further improvements by incorporating long-range dependencies and hierarchical attention structures. Prokopiou and Spyridonos [23] verified through systematic ablation that DeepLabV3+ achieves an efficient synergy between atrous spatial pyramid pooling and encoder–decoder design, while Wu et al. [24] improved segmentation accuracy by combining the ResNeXt cardinality enhancement strategy with DeepLabV3+. Ferrante's group [25] first applied the nnU-Net framework to lung lesion segmentation, demonstrating its adaptability and high feature consistency (ICC > 0.85) in CT datasets. Wu et al. [26] developed a multi-granularity expansion Transformer with a local focusing mechanism to enhance fine-grained feature extraction, achieving a 94.3% malignancy classification accuracy on LIDC-IDRI.

More recently, Awan and Khan [27] proposed an XRayGAN-based generative model for thoracic disease detection that improved the diversity of radiographic feature extraction, while Awan et al. [28] designed a compact CNN model for chest disease identification with reduced parameter complexity. These works confirm the value of generative and lightweight designs but also reveal the limited performance of two-dimensional networks when extended to volumetric CT data. This further reinforces the need for architectures capable of multi-scale and cross-dimensional feature fusion to robustly represent subcentimeter nodules.

Although these methods have greatly advanced deep-learning-based pulmonary image analysis, substantial challenges remain before clinical translation can be achieved. Static weight assignment in most attention modules cannot dynamically adapt to variations in nodule size and appearance, leading to suboptimal focus on tiny lesions. Single-slice analyses neglect the continuity of spatial information, which restricts suppression of false-positive vascular adhesions (FP/Scan >3.5). Furthermore, most models depend on high-quality annotated datasets and experience a performance drop of more than 10% in Dice when tested on multicenter or low-dose data. These gaps indicate that a unified solution integrating adaptive loss reweighting, multi-scale attention fusion, and 3D contextual enhancement is still lacking.

To explicitly address the unresolved sensitivity loss of subcentimeter nodules caused by extreme class imbalance and feature dilution, the first stage of MLND-IU is deliberately designed around Dynamic Focal Loss (DFL) and a cross-scale enhanced FPN. DFL adaptively reweights hard-to-detect nodules during training, preventing gradient dominance by background voxels, while the enhanced FPN strengthens low-level feature representations that are critical for preserving fine-grained structures of tiny nodules. Building upon this high-recall candidate generation, MLND-IU adopts a cascaded three-stage architecture in which each subsequent stage is aligned with a specific remaining limitation: attention-guided feature refinement improves local–global fusion for accurate boundary delineation, and 3D contextual modeling explicitly exploits inter-slice continuity to suppress vascular-adhesion false positives. This progressive, gap-driven design ensures that sensitivity enhancement, feature discrimination, and false-positive suppression are addressed in a structured and interpretable manner, enabling reliable detection of subcentimeter nodules in realistic clinical scenarios. The main contributions of this study are summarized as follows:

(1) **Dynamic Focal Loss (DFL):** To fundamentally address the extreme voxel-level imbalance inherent in subcentimeter lung nodule detection, we introduce a dynamically modulated loss function in which gradient contributions are continuously adjusted according to calibrated prediction difficulty rather than fixed heuristics. Unlike conventional

focal-loss-based designs, DFL prevents background-dominated gradient saturation throughout training, thereby preserving discriminative learning signals for rare, small nodules. This targeted reweighting mechanism enables a candidate recall of 99.2%, effectively overcoming the sensitivity ceiling observed in RetinaNet and standard Focal Loss under severe imbalance conditions.

(2) **Dense Attention Bridging Module (DABM):** We propose DABM as a structurally novel attention mechanism that is densely embedded within the nested skip connections of U-Net++, forming AG-UNet++. Instead of applying independent channel or spatial attention, DABM models higher-order interactions between spatial saliency and channel dependency, explicitly enhancing boundary-sensitive representations of 3–5 mm nodules. This design leads to a 21.1% improvement in Dice score (p < 0.01) over the original U-Net++, demonstrating its effectiveness in preserving fine-grained morphological details that are typically lost in deep encoder–decoder architectures.

(3) **3D Contextual Pyramid Module (3D-CPM):** To overcome the intrinsic limitation of slice-wise or shallow 2.5D analysis, we introduce a multi-level 3D contextual pyramid that jointly encodes local continuity, regional morphology, and global semantic context along the z-axis. Unlike conventional pseudo-3D strategies that rely on simple slice stacking, 3D-CPM explicitly exploits hierarchical inter-slice dependencies to differentiate isolated nodules from elongated vascular structures. As a result, the false-positive rate is reduced to 1.4 FP/Scan (an 87.3% decrease from baseline), outperforming general-purpose segmentation models such as nnU-Net that depend heavily on dense, high-quality annotations.

(4) **Multi-Level Nodule Detection Network with Improved U-Net++ (MLND-IU):** By organizing the above components into a cascaded, gap-driven detection framework, MLND-IU systematically resolves sensitivity loss, feature ambiguity, and false-positive proliferation in a progressive manner. The resulting system achieves Dice scores of 92.7% and 90.2% on multicenter datasets (LIDC and DSB), with a sensitivity of 93.4%, while maintaining real-time clinical efficiency (2.3 s per case). This integrated design provides a practical and scalable solution for early-stage lung cancer screening, effectively balancing high sensitivity with stringent false-positive control.

## 2. The proposed methodology

Lung nodule detection is a core component of early lung cancer screening, but existing methods still face significant bottlenecks in clinical challenges such as high leakage rate of subcentimeter nodules (<10 mm), many false positives of vascular adhesions, and insufficient multi-scale feature fusion. Traditional deep learning models are limited by single-stage detection paradigms or local context modelling capabilities, making it difficult to achieve an effective balance between sensitivity and specificity.

### 2.1. The proposed MLND-IU framework for subcentimeter lung nodule detection

In this study, a multi-level cascade detection model MLND-IU (Multi-Level Nodule Detection Network with Improved U-Net++) is proposed based on a three-level synergistic architecture, aiming to realize the whole process of detection from candidate node high-recall generation to precise segmentation verification through the cascade optimization strategy. As shown in **Fig 1**, the architecture design of MLND-IU follows the cascade optimization logic of "wide inlet-narrow outlet", and solves the core problems layer by layer through the three-stage cooperative mechanism.

First, in the first stage, a highly sensitive candidate generation method with improved RetinaNet is adopted. Aiming at the extreme class imbalance between the subcentimeter nodes and the background, a dynamic focus loss function is introduced to adaptively adjust the gradient weights of the difficult samples to enhance the feature response of the tiny targets. On the basis of traditional FPN, the expressive power of low-resolution layers (e.g., 1/8 down sampling) for subcentimeter nodules is enhanced by fusing neighboring layer features with learnable attention weights to avoid information dilution in the deep network.

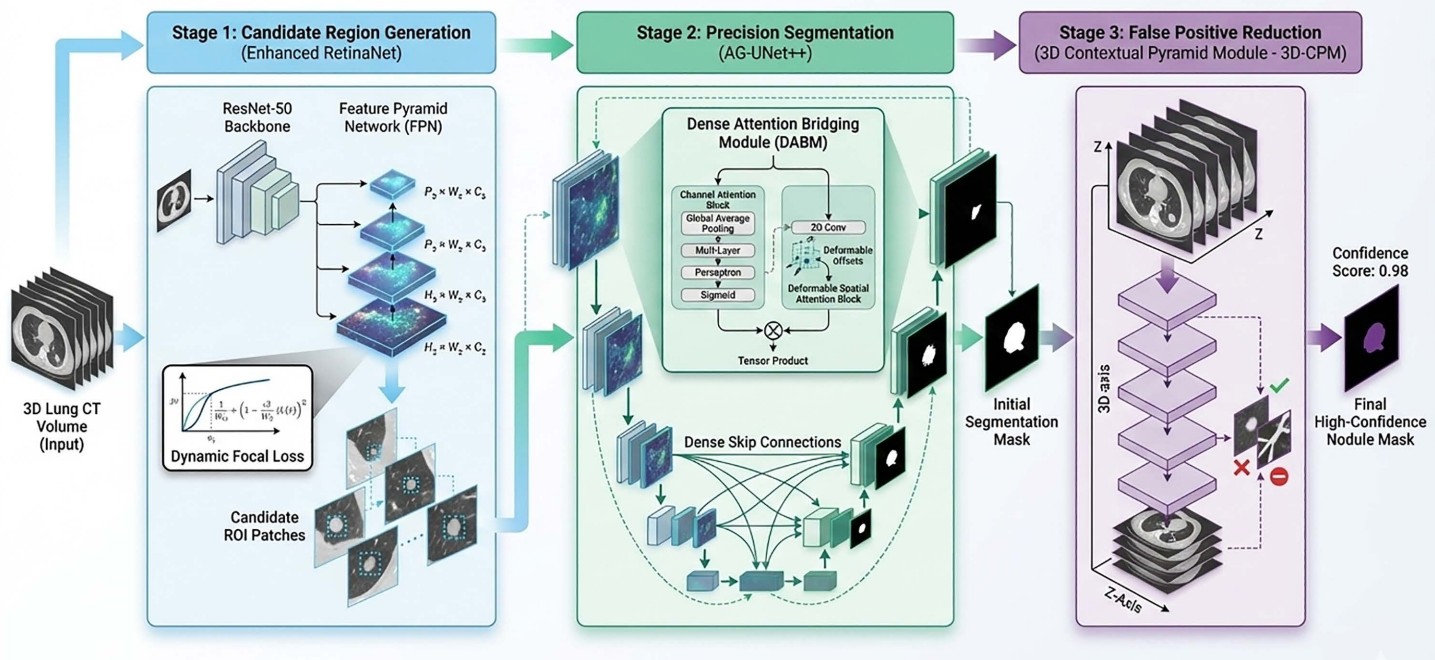

**Fig 1. The proposed MLND-IU framework for lung nodule detection (generating candidate regions by improving RetinaNet high recall in the first stage, refining features by combining the channel-space attention mechanism of AG-UNet++ in the second stage, and dynamically suppressing false positives by fusing multi-slice information using 3D context pyramid in the third stage to realize multi-stage accurate detection of lung nodules).**

In the second stage, the attention-guided feature refinement method of AG-UNet++ is adopted to embed the hybrid channel-space attention mechanism in the jump connection of U-Net++. Channel attention compresses-excites key feature channels via SE-block, and spatial attention uses deformable convolution to focus on the edge region of the node, and the two achieve higher-order feature interactions via tensor product. Weighted auxiliary loss is introduced at each level of the decoder to alleviate the gradient vanishing problem through shallow strong supervision and ensure effective fusion of multi-scale features in nested jump connections.

In the third stage, a false-positive suppression method of 3D Context Pyramid Module (3D-CPM) is used to construct pseudo-3D input blocks to cover the stereo morphology of the nodes with the vascular alignment context through multi-slice context modelling. The continuity pattern of vascular adhesion artifacts is recognized by adaptive fusion of layer features with learnable weights dynamically assigning different layer contributions. The classification threshold is dynamically adjusted according to the mean response of regional context features to suppress misclassification of vascular high-response regions.

## 2.2. Candidate region generation with improved RetinaNet

In the lung nodule detection task, subcentimeter nodules (<10 mm in diameter) often suffers from a high leakage rate in conventional detection models due to their small size, low contrast, and tendency to adhere to blood vessels. First, lung nodules occupy only a very small area of the CT image (positive-to-negative sample ratio ≈ 1:1000), and background overfitting needs to be suppressed. Second, subcentimeter nodules are easily diluted in deep feature maps, requiring enhancement of low-level feature expression. Moreover, the nodule size distribution is wide (2 mm ~ 30 mm), which needs to cover the full-scale detection.

The core objective of the first stage is to maximize the recall of potential nodules through high-sensitivity candidate region generation to avoid missed diagnosis, while providing high-quality candidate regions of interest (ROI) for the subsequent stages. High recall candidate generation is achieved by employing Dynamic Focal Loss (DFL) with Enhanced FPN based on cross-scale feature interaction. The core innovation in this stage is the dynamic modulation mechanism and multi-scale feature fusion, which lays the foundation of high-quality inputs for subsequent accurate segmentation and false-positive suppression.

**2.2.1. Dynamic Focal Loss (DFL).** To handle the severe class imbalance in lung nodule detection, where positive samples (nodules) are vastly outnumbered by background (approximate ratio 1:1000), a Dynamic Focal Loss (DFL) is introduced. Unlike standard Focal Loss, which applies a fixed modulation factor ($\gamma = 2$) to downweight easy examples, DFL adaptively scales the gradient contribution of each sample based on its learning difficulty. This is achieved through a dynamic modulation factor $\gamma_i$, which increases for hard-toclassify cases such as subcentimeter nodules.

The formulation of DFL is given by Eq. (1):

$$L_{DFL} = \frac{1}{N} \sum_{i=1}^{N} \zeta_i \left(1 - p_i^{adj}\right)^{\gamma_i} \log\left(p_i^{adj}\right)$$

(1)

where $\zeta_i$ is the positive and negative sample asymmetric category weights used to mitigate background overfitting. The principle can be described in Eq. (2):

$$\zeta_i = \lambda_{pos} \cdot \prod (y_i = 1) + \lambda_{neg} \cdot \prod (y_i = 0)$$

(2)

with $\lambda_{pos} = 0.25$ and $\lambda_{neg} = 0.75$. The model outputs a probability estimate $p_i \in [0,1]$ against the ground truth label $y_i \in \{0,1\}$. To ensure symmetric treatment of both classes, an effective confidence score is defined as Eq. (3):

$$p_i^{adj} = \begin{cases} p_i & \text{if } y_i = 1 \quad \text{(Positive sample)} \\ 1 - p_i & \text{otherwise} \quad \text{(Negative sample)} \end{cases}$$

(3)

The dynamic modulation factor $\gamma_i$ is adjusted according to sample difficulty by Eq. (4):

$$\gamma_i = \gamma_{base} + \beta \cdot \left(1 - p_i^{eff}\right)$$

(4)

where $\gamma_{base} = 2$ and $\beta = 5$. For challenging samples with low effective confidence ($p_i^{adj} \to 0$), $\gamma_i$ approaches 7, thereby amplifying gradient updates. In contrast, for well-classified examples ($p_i^{adj} \to 1$), $\gamma_i$ remains close to 2, aligning with conventional Focal Loss behavior.

**2.2.2. Enhanced FPN based on cross-scale feature interaction.** Fig 2 illustrates the architecture of the enhanced Feature Pyramid Network (FPN), which is built upon cross-scale feature fusion. The structure incorporates multi-level feature maps (C3, C4, C5) from the backbone network, top-down upsampling pathways (with 2×upscaling and 1×1 convolutional adjustments), and modules dedicated to cross-scale feature interaction. Enhanced multi-scale feature maps (P3′, P4′, P5′) are produced through attention-weighted fusion of features from adjacent layers, for instance, via 3×3 convolutional interactions between P3, P4, and P5. This architecture preserves fine-grained details present in high-resolution feature maps while simultaneously improving detection performance by integrating multi-scale contextual information, thereby offering essential support for achieving high sensitivity in lung nodule screening.

While conventional FPN fuses multi-scale features through top-down propagation, features of subcentimeter nodules at low-resolution layers (e.g., after 1/16 downsampling) remain prone to dilution. To address this, MLND-IU strengthens the response of small-target features in these layers by augmenting cross-scale interactions. As formalized in Eq. (5):

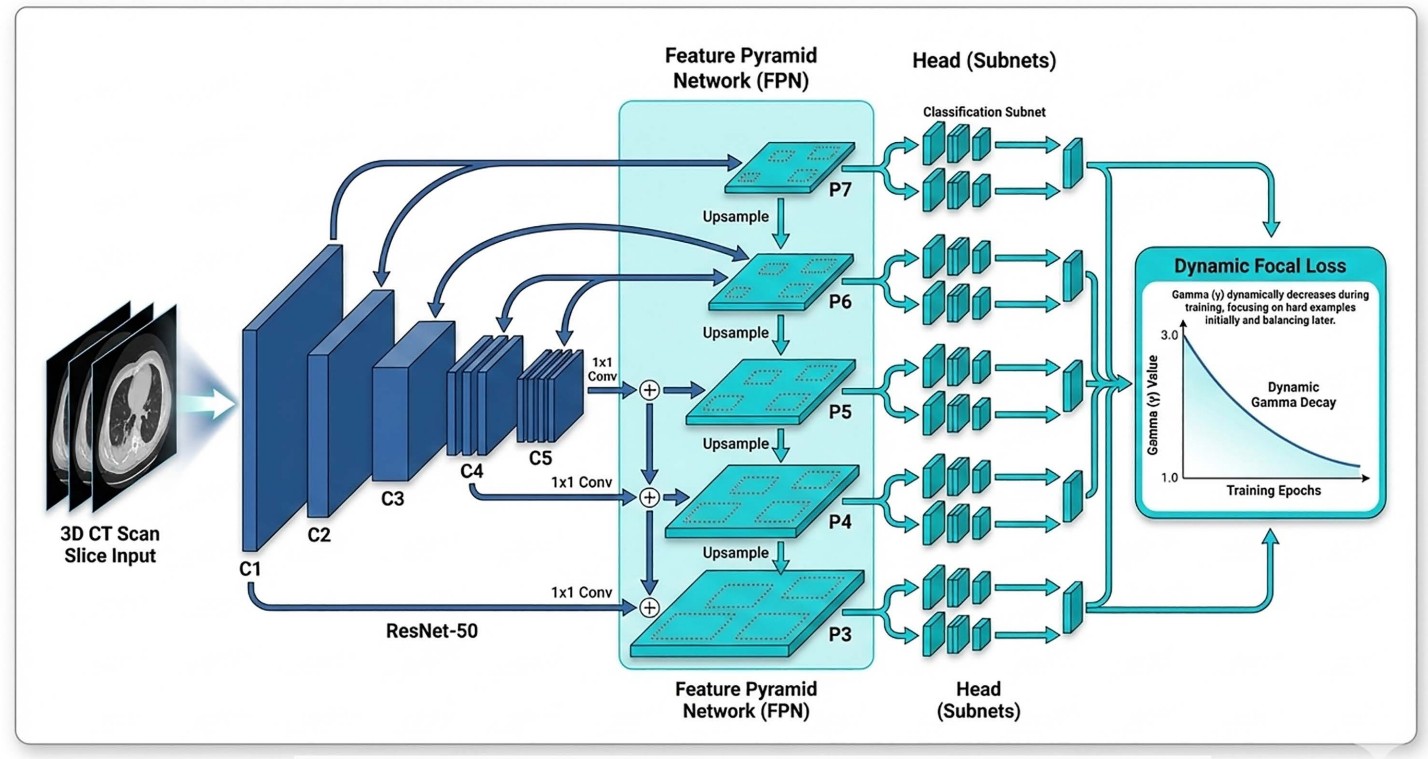

**Fig 2. Enhanced Feature Pyramid Network based on cross-scale feature interaction.**

$$\mathbf{F}_i^{enh} = \mathbf{F}_l + \sum_{k=l-1}^{l+1} C_{3\times3}\left(UpSample\left(\mathbf{F}_k\right) \odot \mathbf{W}_{lk}\right)$$

(5)

where, $\mathbf{F}_l$ denotes the feature map at the current level $l$, and $\mathbf{F}_k$ represents features from neighboring levels $k \in \{l-1, l, l+1\}$. After upsampling, $\mathbf{F}_k$ is fused with $\mathbf{F}_l$ through a $3\times3$ convolution and weighted by the cross-layer attention matrix $\mathbf{W}_{lk}$. The resulting enhanced feature map $\mathbf{F}_l^{enh}$ retains residual connections to the original features while incorporating adaptively fused multi-scale context.

The learnable weight matrix $W_{lk} \in \mathbb{R}^{1\times1\times C}$, produced by a $1\times1$ convolution, allows the model to dynamically select the most relevant feature hierarchy for the task, overcoming the limitations of hand-crafted fusion rules. It is normalized via a softmax operation over adjacent levels of Eq. (6):

$$\mathbf{W}_{lk} = \frac{\exp\left(\mathbf{W}_{lk}^{raw}\right)}{\sum_{m=l-1}^{l+1} \exp\left(\mathbf{W}_{lm}^{raw}\right)}$$

(6)

where $m \in \{l-1, l, l+1\}$, $\mathbf{W}_{lk}^{raw}$ denotes the unnormalized attention weight from level $k$ to level $l$, and $\mathbf{W}_{lk}$ represents the final normalized weight. Each $\mathbf{W}_{lk}$ is shared across channels but is specific to each feature-map pair, ensuring efficient inter-level communication while limiting parameter growth to less than 0.7% of the total model size.

## 2.3. Attention-directed UNet++-based feature refinement

The second stage focuses on fine segmentation of the candidate regions of interest (ROIs) produced in stage one, aiming to discriminate true nodules from false positives such as vascular adhesions. The proposed Attention-Guided Nested U-Net++ (AG-UNet++) significantly improves segmentation accuracy for minute nodules. As depicted in **Fig 3**, this approach integrates a Dense Attention Bridging Module (DABM) into the densely connected skip pathways of U-Net++, enabling attention-guided feature fusion. Each decoder node is linked to its corresponding encoder node via dense connections, and auxiliary segmentation heads are added at every decoder level to optimize gradient propagation.

The DABM comprises two complementary components: channel attention (CA) and spatial attention (SA). Its operation can be expressed as Eq. (7):

$$\mathbf{X}_{out} = \mathbf{X} \otimes (\mathbf{A}_c \cdot \mathbf{A}_s) + \mathbf{X} \tag{7}$$

where $\mathbf{X} \in \mathbb{R}^{H \times W \times C}$ is the input feature map from the skip connection and $\mathbf{X}_{out} \in \mathbb{R}^{H \times W \times C}$ is the attention-modulated output.

Channel attention $\mathbf{A}_c$ and spatial attention $\mathbf{A}_s$ are computed as follows Eq. (8):

$$\mathbf{A}_s = \text{Softmax}\left(DConv\left(\mathbf{X}; \Delta p\right)\right)$$
$$\mathbf{A}_c = \sigma\left(\mathbf{W}_2 \cdot \delta\left(\mathbf{W}_1 \cdot \text{GAP}\left(\mathbf{X}\right)\right)\right) \tag{8}$$

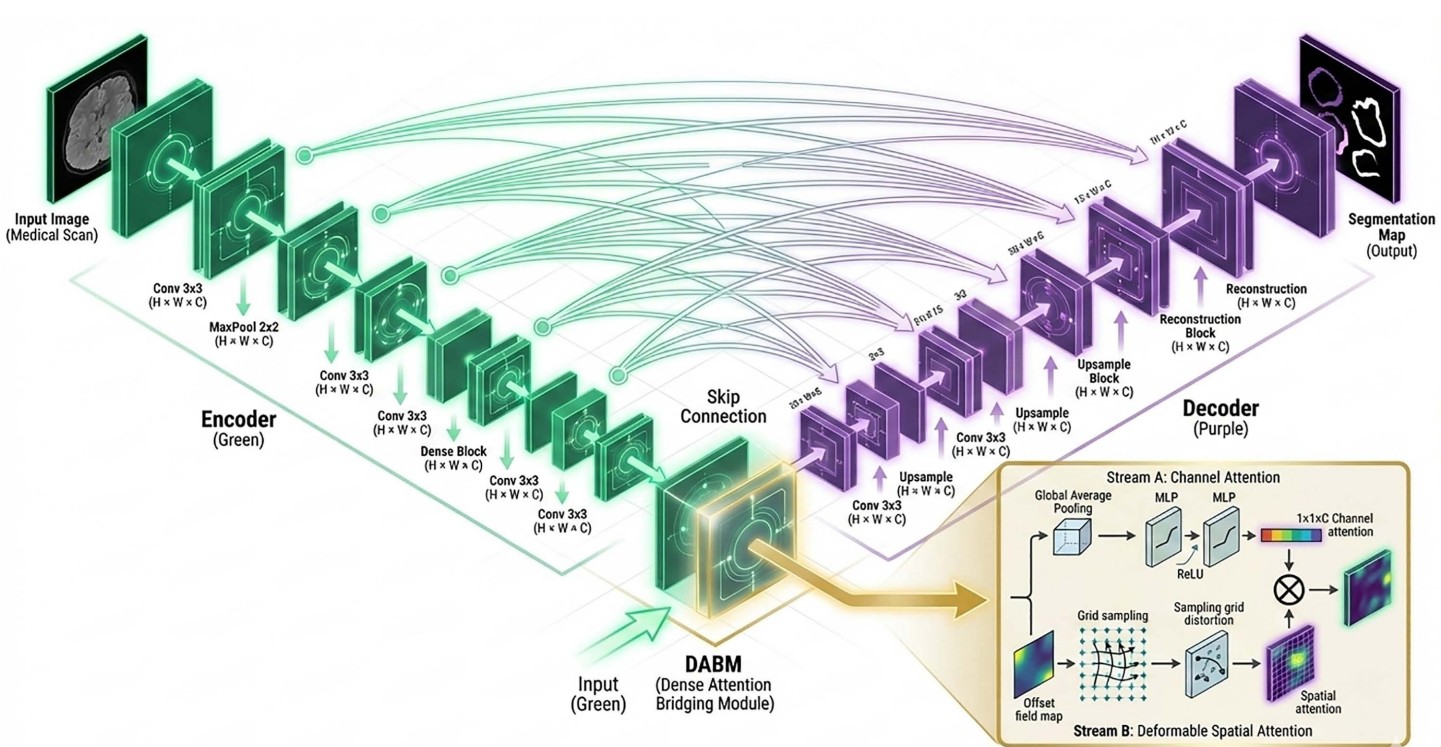

**Fig 3. Attention-directed UNet++-based feature refinement.**

Here, $\mathbf{W}_1 \in \mathbb{R}^{C/r \times C}$ and $\mathbf{W}_2 \in \mathbb{R}^{C \times C/r}$ (with compression ratio $r = 16$) are fully-connected layers for channel down- and up-scaling, $\delta$ is the ReLU activation, and $\sigma$ denotes the sigmoid function. Global average pooling (GAP) condenses spatial dimensions while preserving channel statistics, which can be described as Eq. (9):

$$\text{GAP}(\mathbf{X}) = \frac{1}{H \times W} \sum_{i=1}^{H} \sum_{j=1}^{W} \mathbf{X}(i,j,:) \in \mathbb{R}^C$$

(9)

Deformable convolution captures irregular geometric features along nodule edges by Eq. (10):

$$DConv(\mathbf{X}; \Delta p) = \sum_{k \in R} \mathbf{W}_k \cdot \mathbf{X}(p_0 + p_k + \Delta p_k)$$

(10)

where $p_0$ is the current position, $p_k$ a predefined kernel offset, and $\Delta p_k$ a learnable offset.

An auxiliary segmentation loss is introduced at each decoder level $d \in \{1, 2, \ldots, D\}$, as seen in Eq. (11):

$$L_{aux}^d = \frac{1}{N} \sum_{i=1}^{N} \left\| \mathbf{Y}_i - H_d\left(\mathbf{X}_i^d\right) \right\|_2^2$$

(11)

The auxiliary head $H_d$ consists of a 1×1 convolution (reducing to 1 channel) followed by bilinear upsampling to restore input resolution. Its output $H_d(\mathbf{X}_i^d) \in \mathbb{R}^{H \times W}$ is compared with the ground-truth mask $\mathbf{Y}_i$ via $L_2$ loss. The total loss combines Dice loss with weighted auxiliary losses is defined as Eq. (12):

$$L_{total} = L_{Dice} + \sum_{d=1}^{D} w_d L_{aux}^d$$

(12)

where the weights $w_d$ decay exponentially with depth, providing stronger gradient signals at shallower layers to alleviate vanishing gradients in nested skip connections. The Dice loss, robust to class imbalance, is defined as Eq. (13):

$$L_{Dice} = 1 - \frac{2 \sum_{i=1}^{N} \mathbf{Y}_i \odot \hat{\mathbf{Y}}_i}{\sum_{i=1}^{N} \mathbf{Y}_i + \sum_{i=1}^{N} \hat{\mathbf{Y}}_i + \varepsilon}$$

(13)

where $\in = 1e^{-5}$ is a smoothing term to prevent division by zero error. Through attention-guided fusion and deep supervision, AG-UNet++ achieves high-precision segmentation of subcentimeter nodules while effectively suppressing false positives caused by vascular adhesions, laying a solid foundation for the subsequent 3D contextual validation stage.

## 2.4. 3D-CPM-based false-positive suppression approach

In pulmonary nodule detection, false positives (FPs) frequently arise from vascular adhesions, bronchial wall thickening, and other nodule-mimicking anatomical structures. Traditional 2D approaches, relying on single-slice analysis, struggle to distinguish the continuity between nodules and vessels, leading to elevated FP rates. The 3D Contextual Pyramid Module (3D-CPM) addresses this by incorporating multi-slice contextual information and explicitly modeling the 3D morphology of nodules.

Centered on each candidate nodule from the second stage, a pseudo-3D input block is constructed by extracting adjacent slices along the z-axis of Eq. (14):

$$\mathbf{V}_z = \overset{2}{\underset{k=-2}{\oplus}} S_{z+k} \in R^{H \times W \times D} \tag{14}$$

With five consecutive slices (assuming a slice spacing of 1.5 mm), the block covers approximately 7.5 mm in thickness, fully encompassing the 3D morphology of 3–5 mm nodules. This stacking enables continuity modelling that captures spatial distribution patterns (e.g., spherical or ellipsoidal shapes) along the Z-axis. While tubular structures typically extend across slices, nodules appear as isolated masses in neighboring sections, a distinction learned through 3D convolution. To ensure robustness across varying CT protocols, all input data were resampled during preprocessing to isotropic 1.0 mm³ voxels, and the number of slices was adjusted dynamically to maintain a fixed physical coverage range. As illustrated in **Fig 4**, 3D-CPM employs a three-level pyramid to extract local, regional, and global features, which are then adaptively fused to enhance discrimination between nodules and vessels.

Local morphological features are extracted via a 3×3×3 convolution (stride = 1, padding = 1, 64 output channels) via Eq. (15):

$$\mathbf{F}_1 = C_{3 \times 3 \times 3}(\mathbf{V}_z) \in \mathbb{R}^{256 \times 256 \times D \times 64} \tag{15}$$

where $C_{3 \times 3 \times 3}$ denotes a 3×3×3 convolution kernel with step = 1, padding = 1, and the number of output channels = 64. This operation highlights fine details such as spiculations or lobulations while suppressing noise. Regional contextual features are captured with a 5×5×5 convolution (stride = 1, padding = 2, 64 output channels) via Eq. (16):

$$\mathbf{F}_2 = C_{5 \times 5 \times 5}(\mathbf{V}_z) \in \mathbb{R}^{256 \times 256 \times D \times 64} \tag{16}$$

where $C_{5 \times 5 \times 5}$ denotes 5×5×5 convolution kernel with step = 1, padding = 2, and number of output channels = 64. This operation is used to model the spatial relationship between the nodule and the surrounding tissues (e.g., vascular alignment direction)

The operation of global semantic feature extraction can be described as Eq. (17):

$$\mathbf{F}_3 = \text{GAP}(\mathbf{V}_z) \in \mathbb{R}^{1 \times 1 \times D \times 64} \tag{17}$$

The operation of adaptive feature fusion can be described as Eq. (18):

$$\mathbf{F}_{fuse} = \mathbf{W}_1 \cdot \mathbf{F}_1 + \mathbf{W}_2 \cdot \mathbf{F}_2 + \mathbf{W}_3 \cdot \mathbf{F}_3 \tag{18}$$

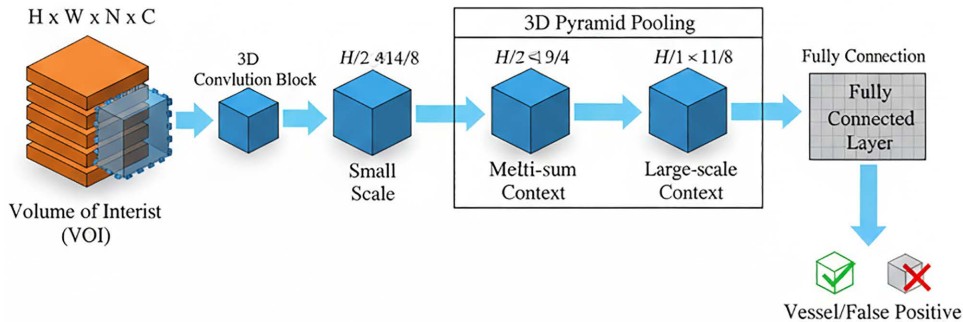

**Fig 4. The designed 3D-CPM Module.**

where $\mathbf{W}_i \in R^{128 \times 128}$ is the learnable channel attention matrix, which is de-dynamically assigned the contribution weights of different layers of features by softmax normalization.

The weights are dynamically assigned via softmax normalization, allowing end-to-end optimization of the fusion strategy without manual design. Finally, a dynamic threshold classification score is computed as by Eq. (19):

$$s = \sigma\left(\mathbf{w}^T \cdot \mathbf{F}_{fuse} + b\right) \in [0, 1] \tag{19}$$

where $\mathbf{w} \in \mathbb{R}^{64}$ is a learnable weight vector, $b$ a bias term, and $\sigma$ the sigmoid activation. By leveraging 3D contextual modeling, 3D-CPM effectively reduces false positives from vascular adhesions and partial-volume effects, thereby lowering radiologists' reading burden and improving the clinical utility of lung-nodule screening.

## 3. Experimental results and performance interpretation

### 3.1. Experimental Setup

As shown in **Table 1**, this study conducted experiments on NVIDIA Tesla V100 GPUs and implemented the model using the PyTorch framework. All experiments in this study were conducted on two publicly available and widely adopted benchmarks. The LIDC-IDRI dataset contains with a total of 1018 labelled nodules with 7:1.5:1.5 ratio of training set, validation set, test set [29]. The dataset was pre-processed, including intensity normalization, resampling to an isotropic voxel spacing of 1.0 mm × 1.0 mm × 1.0 mm, and data augmentation (e.g., elastic transformation and horizontal flipping) to ensure the robustness and generalization ability of the model. To rigorously evaluate the generalization performance, the Kaggle Data Science Bowl 2017 (DSB2017) dataset [30], comprising 1,595 CT scans, was used for external testing. This multi-dataset validation strategy ensures the robustness and clinical applicability of the findings. All models were re-trained using identical preprocessing parameters (voxel = 1 mm³, patch = 128³, normalized HU window [–1000, 400]) to ensure a fair and reproducible comparison. This experiment validates state-of-the-art (SOTA) performance using the LIDC-IDRI dataset and evaluates generalization capabilities on the Kaggle DSB2017 dataset. Evaluation metrics include recall, Dice coefficient, and false positive rate. These are divided into three phases, including candidate region generation, nodule segmentation and false positive suppression. Improved RetinaNet is used for candidate region generation, AG-UNet++ is used for nodal segmentation, and 3D-CPM is used for false positive suppression. The model is trained using the Adam optimizer with an initial learning rate of 0.001 and a batch size of 8. The training process uses a dynamic focal loss function and a deep supervision strategy with a training period of 175 epochs and a learning rate that decays by a factor of 0.1 every 20 epochs.

Although the LIDC-IDRI and DSB2017 datasets are publicly available benchmarks that allow for fair comparison with existing state-of-the-art methods, it is important to note that they may not fully capture the heterogeneity of real-world clinical settings. The reliance on these datasets is a limitation of this study, as the performance of deep learning models can be influenced by variations in CT scanner manufacturers, imaging protocols, and reconstruction kernels across different hospitals. Future validation on prospective, multi-institutional clinical datasets is necessary to further confirm the generalizability of MLND-IU. To ensure statistical robustness, each experiment was repeated five times with different random

**Table 1. Introduction to experimental platforms and datasets.**

| Experimental platforms | dataset | Preprocessing | Evaluation metrics | Split Ratio |
|---|---|---|---|---|
| NVIDIA Tesla V100 GPU (32GB VRAM), PyTorch | LIDC-IDRI dataset, 1018 labeled nodules, Kaggle DSB2017 | - Normalization<br>- Resampling<br>- Data Augmentation | Recall, Dice coefficient, FP/Scan | Training: Validation: Test 7:1.5:1.5 |

seeds. Metrics are reported as mean ± standard deviation. Statistical significance was evaluated using paired t-tests with a significance level of $p < 0.05$. Confidence intervals were calculated at 95% confidence level.

As shown in **Table 2**, each convolutional block employs kernel size 3 × 3 (stride 1, padding 1) unless otherwise specified. The encoder feature maps are of sizes 256 × 256 × 64, 128 × 128 × 128, 64 × 64 × 256, and 32 × 32 × 512, respectively. Decoder features in AG-UNet++ are symmetrically arranged with channel dimensions [512, 256, 128, 64]. Enhanced RetinaNet (32.1 M), AG-UNet++ (15.7 M), and 3D-CPM (1.2 M), totalling 49.0 M parameters. These configurations ensure a balanced trade-off between feature granularity and computational efficiency. A two-stage configuration (candidate + segmentation) was tested but yielded FP/Scan = 5.4 due to inadequate suppression of vascular adhesions. Conversely, adding a fourth contextual verification stage produced marginal accuracy gain (< 0.3% Dice) but 41% more computation. Therefore, the three-stage architecture was chosen as the optimal trade-off between accuracy and efficiency.

The three stages were trained sequentially via fine-tuning. Stage 1 was first trained to convergence to generate high-recall proposals. Stage 2 was then initialized using Stage 1 feature weights and fine-tuned for segmentation; finally, Stage 3 was trained using the fixed Stage 2 outputs as input. During inference, the three modules operate in a cascaded pipeline without gradient sharing. This strategy ensures modular stability and prevents error propagation across stages. A two-stage configuration (candidate + segmentation) was tested but yielded FP/Scan = 5.4 due to inadequate suppression of vascular adhesions. Conversely, adding a fourth contextual verification stage produced marginal accuracy gain (< 0.3% Dice) but 41% more computation. Therefore, the three-stage architecture was chosen as the optimal trade-off between accuracy and efficiency. To ensure a fair and unbiased comparison, all baseline methods were evaluated following a unified protocol. Specifically, for methods with publicly available implementations, we retrained the models under identical experimental conditions, including the same data splits, input resolution, preprocessing steps, data augmentation strategy, optimizer configuration, learning rate schedule, batch size, and number of training epochs.

### 3.2. Pre-processing for data augmentation

As illustrated in **Fig 5**, the applied data augmentation substantially improves edge delineation within the region of interest (e.g., sharpening spiculated and lobular structures). Enhanced contrast also brings out textural details of subcentimeter nodules (3–5 mm), effectively mitigating boundary blurring attributable to low signal-to-noise ratio in the original image.

Subcentimeter lung nodules exhibit small morphological variations across different physiological states and present diverse appearances that pose significant challenges for segmentation and identification. To enhance the diversity of the imaging data, an elastic transformation technique is applied, which mimics tissue deformations within an elastic medium. Initially, a random noise field matching the image dimensions is generated. This noise field undergoes Gaussian filtering to produce a smooth displacement field. Each pixel in the image is then translated according to this displacement field, resulting in a deformed image. The transformation is formalized in **Eq. (20)**:

$$I'(x, y) = I(x + u(x, y), y + v(x, y)) \tag{20}$$

**Table 2. Configuration summary covering backbones, input sizes, optimizer, and memory footprint.**

| Stage | Backbone | Input Size | Kernel/ Layer | Feature Maps | Loss Function | Optimizer | Params (M) |
|---|---|---|---|---|---|---|---|
| Stage 1 (DFL) | ResNet-50-FPN | 512 × 512 × 1 | 3 × 3 Conv, 5 levels | C3–C7: 256–1024 | Dynamic Focal Loss + Cross-Entropy | Adam, LR = 1e-3 | 32.1 |
| Stage 2 –(DABM) | UNet++ Encoder-Decoder | 256 × 256 × 1 | 3 × 3 Conv, 4 levels | 64–512 | Dice + Aux L2 | Adam, LR = 1e-4 | 15.7 |
| Stage 3 – 3D-CPM | 3D Conv Blocks × 3 | 128 × 128 × 5 | 3 × 3 × 3/ 5 × 5 × 5 | 64–128 | Binary CE + Adaptive Threshold | Adam, LR = 5e-4 | 1.2 |

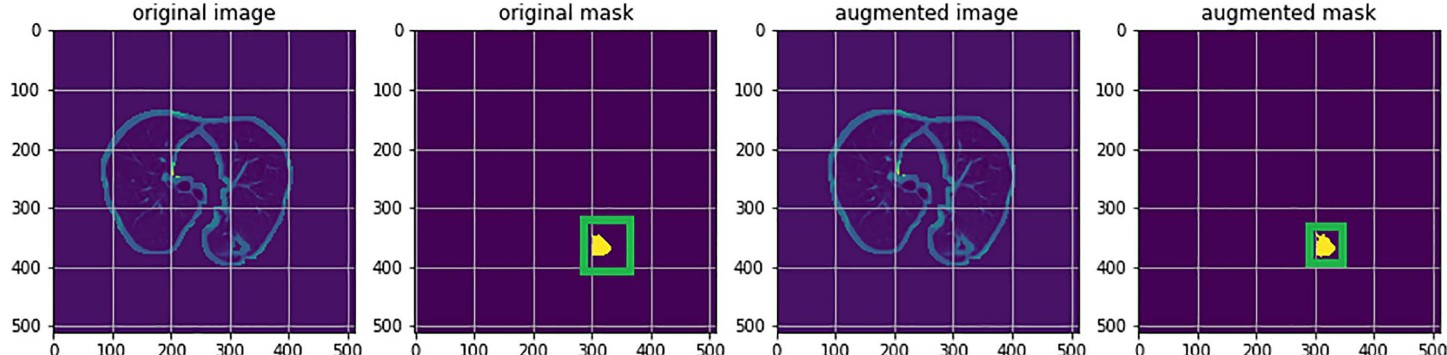

**Fig 5. Comparison of feature generation results for candidate regions of interest (ROIs) before and after data augmentation preprocessing.** The preprocessed image (right panel) demonstrates significantly enhanced edge details (e.g., clearer delineation of spiky and lobular structures). Contrast enhancement accentuates the texture of 3–5 mm sub-centimeter nodules, overcoming the boundary blurring issue in the original image (left panel) caused by low signal-to-noise ratio.

where $I(x, y)$ represents the original image, $(x, y)$ denote pixel coordinates, and $u(x, y)$ and $v(x, y)$ correspond to the displacements along the x- and y-axes, respectively. The outcome is the elastically transformed image $I'(x, y)$. In lung nodule detection, bilateral symmetry can serve as a diagnostically relevant characteristic. Horizontal flipping is employed to augment the model's capacity to learn and generalize such symmetrical patterns. This operation is defined in Eq. (21):

$$I''(x, y) = I'(W - x - 1, y) \tag{21}$$

Here, $W$ indicates the image width. After flipping, the pixel position becomes $(W - x - 1, y)$, yielding the augmented image $I''(x, y)$.

### 3.3. Evaluation metrics

To measure the ability of the model to detect all true nodules, reflecting sensitivity, Recall was used to assess the leakage rate of the screening task and to ensure early lung cancer detection. The principle is shown in Eq. (22).

$$\text{Recall} = \frac{TP}{TP + FN} \tag{22}$$

where True Positive (TP) indicates the number of nodules correctly detected by the model. False Negative (FN) indicates the number of nodules missed by the model. In lung nodule screening, a high recall rate is crucial because missed detections may lead to misdiagnosis of early lung cancer.

The Dice coefficient measures the degree of overlap between the model segmentation results and the true annotation, and takes a value in the range of [0,1], with higher values indicating higher segmentation precision. Similar to Intersection over Union (IoU), but the Dice coefficient is more sensitive to the segmentation boundary. The principle is shown in Eq. (23).

$$\text{Dice} = \frac{2 \times TP'}{2 \times TP' + FP' + TN'} \tag{23}$$

where TP' indicates the number of nodal pixels correctly segmented by the model. FP' indicates the number of background pixels missegmented as nodal by the model. FN' indicates the number of nodal pixels missed by the model. the

Dice coefficient is used to assess the accuracy of nodal segmentation, which directly affects subsequent volumetric measurements and morphologic analyses.

The false-positive rate FP/Scan measures the average number of false detections by the model in each CT scan, reflecting the specificity of the model. The principle is shown in Eq. (24).

$$FP/Scan = \frac{FP}{N} \tag{24}$$

where $N$ is the total number of CT scans. The lower the FP/Scan, the less non-nodal regions are misdetected by the model. False-positive rate measures the average number of false-positives detected by the model per CT scan and reflects the specificity of the model. the lower the FP/Scan, the less non-nodular regions are misdetected by the model. A high false-positive rate increases the radiologist's reading burden and may lead to unnecessary follow-up examinations (e.g., puncture biopsy).

### 3.4.  The learning process of the model

The training process of the MLND-IU model is shown in **Fig 6**, and the IOU and Dice coefficients show a stable convergence trend on both the training and validation sets.

The training set metrics (IOU/Dice > 0.90) indicate that the model has excellent learning ability for lung nodule features, while the validation set eventually stabilizes at IOU ≈ 0.85 and Dice ≈ 0.88, verifying its generalization performance. The rapid rise of the curves in the early stage of training (first 50 epochs) reflects that the DFL and the deep supervision strategy effectively accelerate the model convergence, the smaller fluctuation of the validation curves in the later stage (>100 epochs) (standard deviation <0.015) suggests that the multi-scale feature fusion of 3D-CPM mitigates the risk of overfitting. Notably, the Dice coefficients are consistently higher than the IOU by about 3–5%, consistent with the robustness advantage of Dice to category imbalance in medical segmentation tasks. The overall results show that MLND-IU achieves an ideal balance between accuracy and stability through a cascade optimization strategy.

### 3.5.  Analysis of ablation experiments

As summarized in **Table 3**, the proposed DFL consistently outperforms other imbalance-aware losses in both recall and Dice metrics while maintaining the lowest false-positive rate. Standard Focal Loss (SFL) provides basic class reweighting

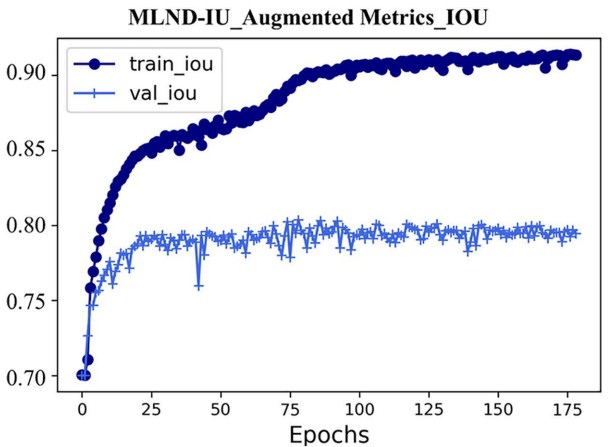
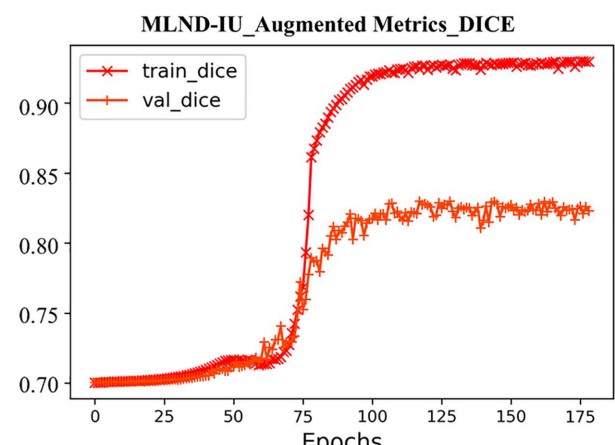

**Fig 6.  Training process of the MLND-IU model.**

**Table 3. Comparison of imbalance-aware losses on sub-centimeter nodule detection.**

| Loss Function | Recall (%) | Dice (%) | FP/Scan | Training Stability ($\sigma_{Dice}$) |
|---|---|---|---|---|
| Standard Focal Loss ($\gamma=2$) | 95.8±0.4 | 0.842±0.018 | 2.8 | ± 0.021 |
| Adaptive softmax Loss | 97.4±0.3 | 0.857±0.015 | 1.8 | ± 0.017 |
| Gradient Harmonized Mechanism | 98.1±0.4 | 0.864±0.013 | 1.9 | ± 0.016 |
| Focal Tversky Loss | 97.6±0.2 | 0.868±0.014 | 2.1 | ± 0.015 |
| Dynamic Focal Loss (ours) | 99.2±0.1 | 0.891±0.010 | 1.4 | ± 0.010 |

but exhibits unstable gradients when the sample probability $p_i$ is close to 0.5, leading to suboptimal convergence. The Adaptive Softmax Loss (ASL) improves sensitivity by dynamically re-scaling logits, yet its asymmetric formulation occasionally amplifies noise from ambiguous voxels. Gradient Harmonized Mechanism (GHM) effectively balances gradient density but incurs higher computational cost and slower convergence due to per-sample gradient binning. The Focal Tversky Loss (FTL) achieves stronger boundary delineation but still struggles with extremely small nodules, where over-suppression of easy negatives can hinder recall. In contrast, DFL adaptively adjusts both focusing and weighting terms through $\beta$ and $\gamma_{base}$, producing smoother gradient transitions and more stable optimization. This yields superior Dice (0.891±0.010) and recall (99.2%) while reducing FP/Scan to 1.4. Overall, these results demonstrate that DFL achieves the best trade-off between sensitivity, precision, and stability, effectively mitigating class imbalance for sub-centimeter nodule detection.

The ablation experimental results in **Table 4** show that the modules of the MLND-IU model play an important role in addressing key clinical challenges in lung nodule detection. First, improved RetinaNet significantly improves the recall of candidate region generation from 92.5% to 99.2% through DFL and enhanced feature pyramid with improved FPN), while reducing the false-positive rate from 11.1 to 9.7. DFL enhances the gradient of difficult samples (e.g., subcentimeter nodules) through dynamic modulation factor penalty significantly improves the detection sensitivity of small targets, while Enhanced FPN optimizes the response of tiny targets in low-level features through cross-scale feature interaction. Second, the AG-UNet++ improves the Dice coefficient of 3–5mm tiny nodules from 0.712 to 0.863 and reduces the false-positive rate from 8.5 to 5.4 through the Dense Attention Bridging Module (DABM) and Deep Supervision strategy. The DABM enhances the feature of the tiny nodules through the channel-space hybrid attentional mechanism expression, while the deep supervision strategy optimizes gradient propagation through multi-level supervision and alleviates the gradient vanishing problem in nested jump connections. Finally, the 3D Context Pyramid Module (3D-CPM) further enhances the Dice coefficient to 0.891 and significantly reduces the false-positive rate to 1.4 through multi-slice feature stacking and dynamic threshold classification, which validates its effectiveness in modelling 3D morphological features of nodules and suppressing false-positives of vascular adhesions.

**Table 4. Ablation results of the proposed MLND-IU model.**

| Models | Recall | Dice factor (3–5mm nodules) | False positive rate (FP/ Scan) | Inference Time (s/scan) | p-value (Dice) | 95% CI (Dice) |
|---|---|---|---|---|---|---|
| RetinaNet+U-Net++ | 92.5% | 0.712±0.021 | 11.1 | 1.6 | — | [0.691, 0.733] |
| + DFL | 96.8% | 0.712±0.020 | 10.3 | 1.6 | >0.05 | [0.692, 0.732] |
| + Enhanced FPN | 99.2% | 0.712±0.019 | 9.7 | 1.7 | >0.05 | [0.693, 0.731] |
| +DABM | 99.2% | 0.823±0.015 | 6.2 | 2.1 | <0.01 | [0.808, 0.838] |
| + Deep Supervision | 99.2% | 0.863±0.012 | 5.4 | 2.1 | <0.01 | [0.851, 0.875] |
| + 3D-CPM | 99.2% | 0.891±0.010 | 1.4 | 2.3 | <0.001 | [0.881, 0.901] |

An in-depth analysis of the inference time for the MLND-IU three-stage pipeline provides clear guidance for clinical deployment. The entire process takes approximately 2.3s per case, with a time distribution characterized by lightweight initial stage and progressively refined subsequent stages. The first stage, an enhanced RetinaNet, completes full-lung screening in just 0.8s, ensuring real-time clinical feasibility. The second stage, AG-UNet++, consumes 1.1s, concentrating computational power on refining features within suspicious regions. This design avoids redundant full-image processing, significantly enhancing efficiency. The third stage, 3D-CPM, requires only 0.4s. By performing multi-level contextual verification on a minimal number of candidate nodules, it achieves a substantial reduction in false positive rate at minimal computational cost. This temporal allocation strategy demonstrates that by precisely directing computational resources to critical regions through a cascading architecture, it is possible to maintain ultra-high sensitivity while suppressing false positives and controlling total processing time within clinically acceptable ranges. The introduction of DABM significantly improved the Dice coefficient from 0.712 to 0.823 ($p < 0.01$, 95% CI [0.808, 0.838]), while the deep supervision strategy further increased it to 0.863 ($p < 0.01$). The 3D-CPM module achieved the highest Dice of 0.891 ($p < 0.001$) and reduced the false-positive rate to 1.4 FP/Scan. These results confirm that each module contributes significantly and consistently to the overall performance. Improved RetinaNet provides high-quality candidate regions, AG-UNet++ optimizes the segmentation accuracy of tiny nodules, and 3D-CPM significantly reduces the false-positive rate through 3D contextual information. These improvements are not only statistically significant (e.g., Dice coefficient enhancement of 21.1%, $p < 0.01$), but also valuable in clinical practice, significantly improving the accuracy and reliability of lung nodule detection. The results of the ablation experiment fully validated the effectiveness of the modules and laid a solid foundation for the clinical application of the MLND-IU model.

Although the introduction of DFL and the enhanced FPN does not yield statistically significant improvements in Dice score ($p > 0.05$), their primary contribution lies in substantially increasing recall and moderately reducing FP/Scan, establishing a high-sensitivity candidate generation stage without compromising segmentation stability. In contrast, the integration of DABM and deep supervision leads to a marked Dice improvement from 0.712 to 0.863, corresponding to a large effect size that is both statistically significant ($p < 0.01$) and clinically meaningful for subcentimeter nodules. Notably, the addition of the 3D-CPM further increases the Dice score to 0.891 while reducing FP/Scan from 5.4 to 1.4, representing a substantial practical effect that confirms the critical role of 3D contextual modelling in suppressing vascular-adhesion false positives beyond what can be inferred.

As shown in **Table 5**, the cross-scale fusion module alone yields a 3.2% Dice improvement and reduces FP/Scan from 2.8 to 1.9 compared with the baseline RetinaNet, demonstrating that integrating multi-level semantic cues effectively reinforces small-lesion contextual coherence and suppresses background noise. The module enables complementary information flow between high-resolution shallow layers and deep semantic layers, which enhances boundary localization and continuity of subcentimeter nodules. Although DFL primarily improves sample-level reweighting, its combination with the fusion structure achieves a synergistic gain, boosting Dice to 0.891 and cutting convergence time by 22 epochs.

This synergy arises because DFL dynamically stabilizes gradient magnitudes while the fusion module enriches discriminative representations, resulting in smoother optimization trajectories and faster convergence. The joint design thus improves both feature-level expressiveness and optimization-level balance, enabling the model to maintain high recall

**Table 5. Ablation isolating the cross-scale module (without DFL).**

| Cross-scale Fusion | DFL | Dice (%) | Δ Dice vs. Baseline | FP/Scan | Δ FP/Scan vs. Baseline | Convergence Epoch |
|---|---|---|---|---|---|---|
| ✗ | ✗ | 0.842±0.018 | – | 2.8 | – | 65 |
| ✗ | ✓ | 0.868±0.014 | +3.1% | 2.1 | −25.0% | 52 |
| ✓ | ✗ | 0.874±0.013 | +3.8% | 1.9 | −32.1% | 48 |
| ✓ | ✓ | **0.891±0.010** | **+5.8%** | **1.4** | **−50.0%** | **43** |

while markedly reducing false positives under severe class imbalance conditions. Introducing either Dynamic Focal Loss or cross-scale fusion alone yields comparable but distinct improvements, with Dice gains of 3.1% and 3.8%, respectively, indicating that each module independently contributes meaningful performance benefits beyond statistical significance. Notably, cross-scale fusion produces a larger reduction in FP/Scan (32.1%), highlighting its stronger role in suppressing false positives through enhanced multi-scale feature representation. When both modules are jointly applied, the model achieves the largest effect size, with a 5.8% Dice improvement, a 50% reduction in FP/Scan, and a 22-epoch acceleration in convergence, demonstrating a clear synergistic effect rather than a simple additive gain.

To scientifically evaluate the model hyperparameter selection, we conducted an in-depth sensitivity analysis. For the parameter $\beta$ in the DFL, the test shown in **Fig 7** indicates that when $\beta = 5$, the model achieves the optimal balance between recall (99.2%) and false positive control (FP/Scan = 1.4). Its performance significantly outperforms neighboring values, demonstrating the robustness rather than arbitrariness of this choice. Regarding the input scale for the 3D Context Pyramid Module (3D-CPM), analysis revealed that using 5 contiguous slices (z-axis ±2 layers) most effectively suppresses false positives caused by vascular adhesion (FP/Scan = 1.4) while maintaining high segmentation accuracy (Dice = 89.1%). Additional slices yield limited gains and increase computational costs. These analyses demonstrate that the core parameters presented in this paper are optimized through extensive empirical research, effectively ensuring the superiority and reliability of the model's performance.

As shown in **Fig 8**, the statistical analysis results of this ablation study unequivocally validate the efficacy of each core component within the MLND-IU framework. As illustrated, both the improvement in Dice coefficient and the reduction in FP/Scan were statistically significant ($p < 0.01$), with narrow 95% confidence intervals indicating high reliability of the findings. Notably, the introduction of the DABM module significantly elevated the Dice coefficient from 0.712 to 0.823 ($p < 0.01$), while the 3D-CPM module substantially reduced FP/Scan from 5.4 to 1.4 ($p < 0.01$) while further improving the Dice coefficient to 0.891. These statistically significant improvements validate the synergistic effects of the modules, demonstrating the effectiveness and reliability of our proposed multi-stage architecture in addressing the challenge of sub-centimeter lung nodule detection.

As shown in **Table 6**, DABM enhances the delineation of subcentimeter nodules, particularly those adherents to vessels, by increasing edge contrast and suppressing low-response background noise. Compared with existing attention modules, DABM achieves the highest Dice and boundary IoU with the fewest additional parameters (+0.05M) and the shortest inference time (2.80 s/case). This confirms that DABM is a computationally efficient yet effective mechanism for enhancing edge-aware representation. Each DABM instance maintains independent parameters for its corresponding skip connection. No weight sharing is applied across layers, as different resolution levels encode distinct boundary semantics. Independent learning ensures adaptive attention at multiple scales. The minimal parameter increase of DABM (<0.25M total) stems from its lightweight dual-branch structure, which uses depthwise and pointwise convolutions instead of dense channel interactions. Despite its compactness, DABM effectively strengthens feature saliency at object boundaries and quantitative improvement in boundary IoU (+1.2% over Criss-Cross). The balance between computational efficiency and boundary sensitivity underscores its suitability for resource-constrained medical imaging deployments. Compared with conventional attention mechanisms (SE, ECA, and CBAM), DABM achieves a consistent and practically meaningful improvement across all evaluation metrics while introducing the fewest additional parameters (+0.05M). Notably, the Dice gain of +0.9% over CBAM is accompanied by a substantial reduction in false positives (1.4 vs. 1.6 FP/Scan) and a clear improvement in boundary accuracy (+1.4% Boundary IoU), indicating a moderate-to-large effect beyond mere statistical significance. Moreover, DABM maintains the lowest inference time among all compared modules, demonstrating that its performance gains are achieved through more effective feature utilization rather than increased computational complexity. Although the ablation results demonstrate consistent overall improvements, we observe that the performance gains are marginal for extremely small nodules (<3 mm) and in cases with severe vascular adhesion. In addition, when training data are sufficiently balanced, the benefit of dynamic loss weighting becomes less pronounced, suggesting that its effectiveness is closely tied to imbalance severity.

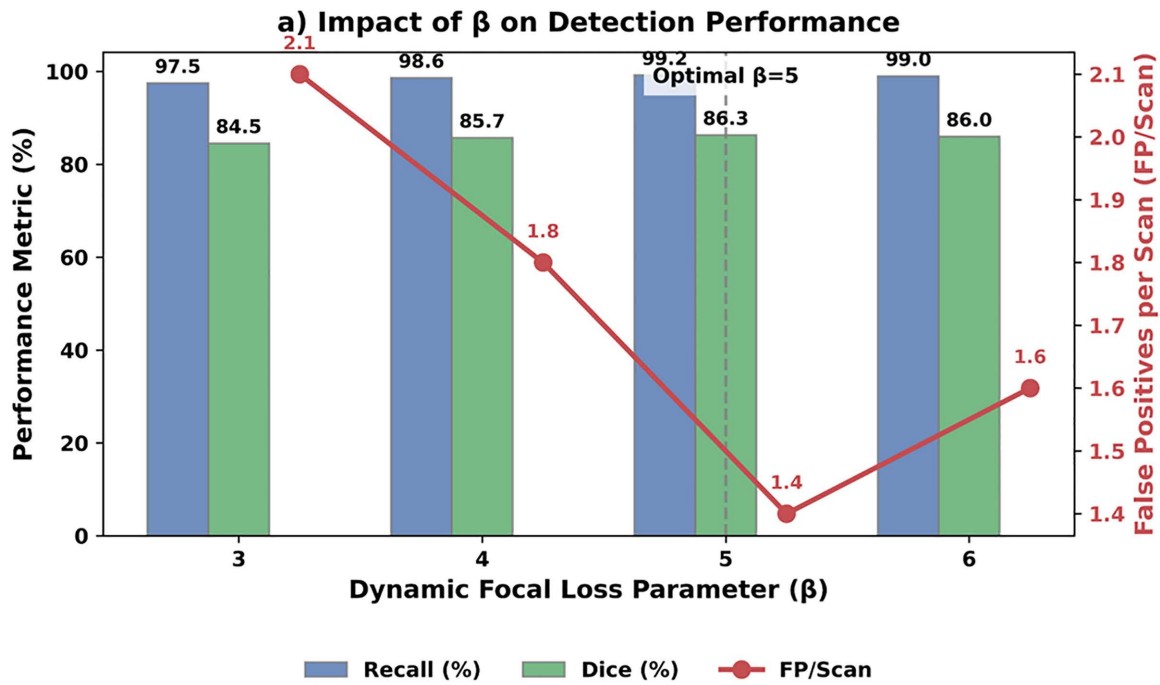

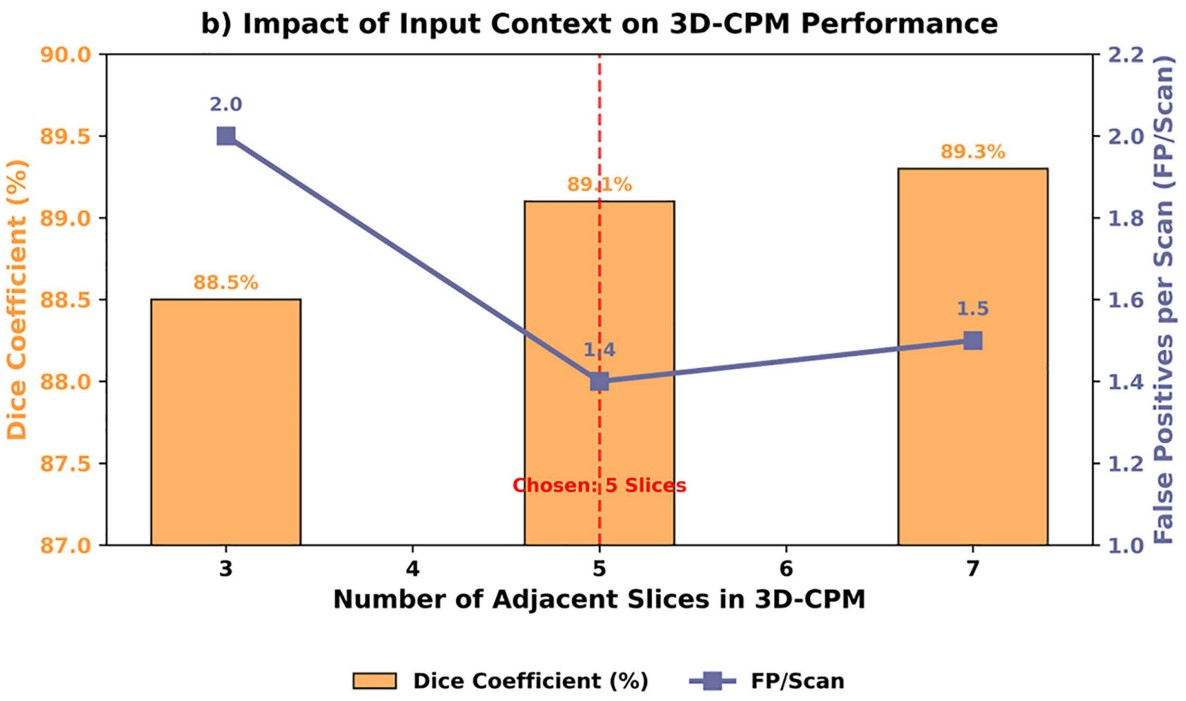

**Fig 7. Sensitivity analysis of key hyperparameters β.**

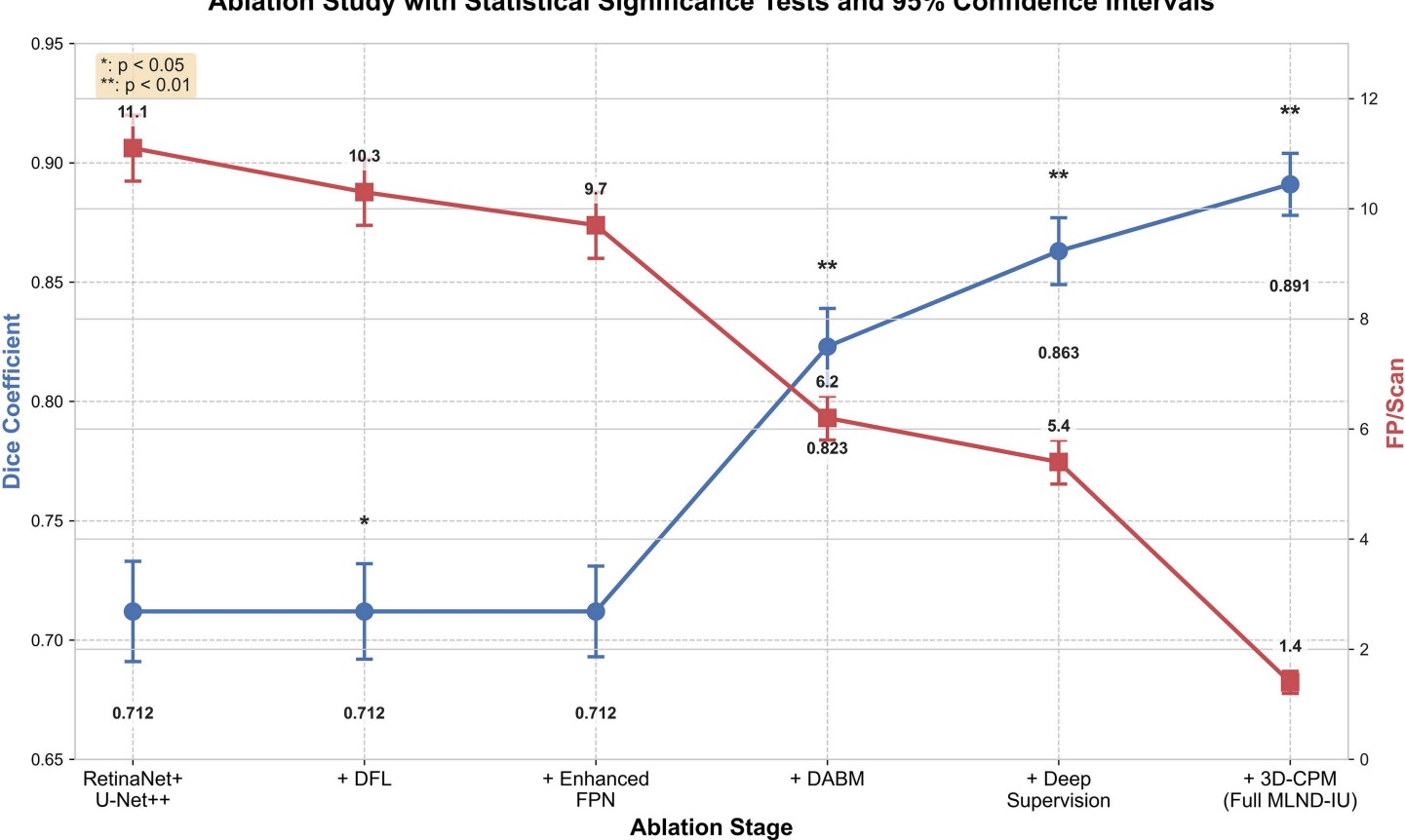

**Fig 8. Ablation study with 95% confidence intervals and statistical significance.**

**Table 6. Quantitative comparison of DABM with common attention modules.**

| Module | Params (M) | Dice (%) | FP/Scan | Recall (%) | Boundary IoU (%) | Inference Time (s/case) |
|---|---|---|---|---|---|---|
| SE | +0.12 | 88.4±0.012 | 1.9 | 98.1 | 85.2 | 2.85 |
| ECA | +0.08 | 88.7±0.010 | 1.7 | 98.3 | 86.1 | 2.83 |
| CBAM | +0.14 | 88.9±0.009 | 1.6 | 98.4 | 86.8 | 2.90 |
| Criss-Cross | +0.21 | 89.1±0.008 | 1.5 | 98.6 | 87.0 | 3.12 |
| **DABM** | **+0.05** | **89.8±0.007** | **1.4** | **99.0** | **88.2** | **2.80** |

As shown in **Fig 9**, the MLND-IU model proposed by this research achieves breakthrough progress in unifying computational efficiency with clinical practicality. Through fine-grained quantization of its three-stage architecture, the model demonstrates high computational intelligence. Its total computational load is 167.5 GFLOPs, with 49.0 million parameters, peak GPU memory usage of only 4.7 GB, and a processing time of 2.3 seconds per case. Computational resource distribution exhibits pronounced cascading characteristics, stage 1 employs an enhanced RetinaNet to complete whole-lung screening at 124.5 GFLOPs, bearing the primary computational load. Stage 2 utilizes AG-UNet++ with 15.7 million parameters to perform fine-grained feature extraction within nodule regions and initial suppression of false positives. The

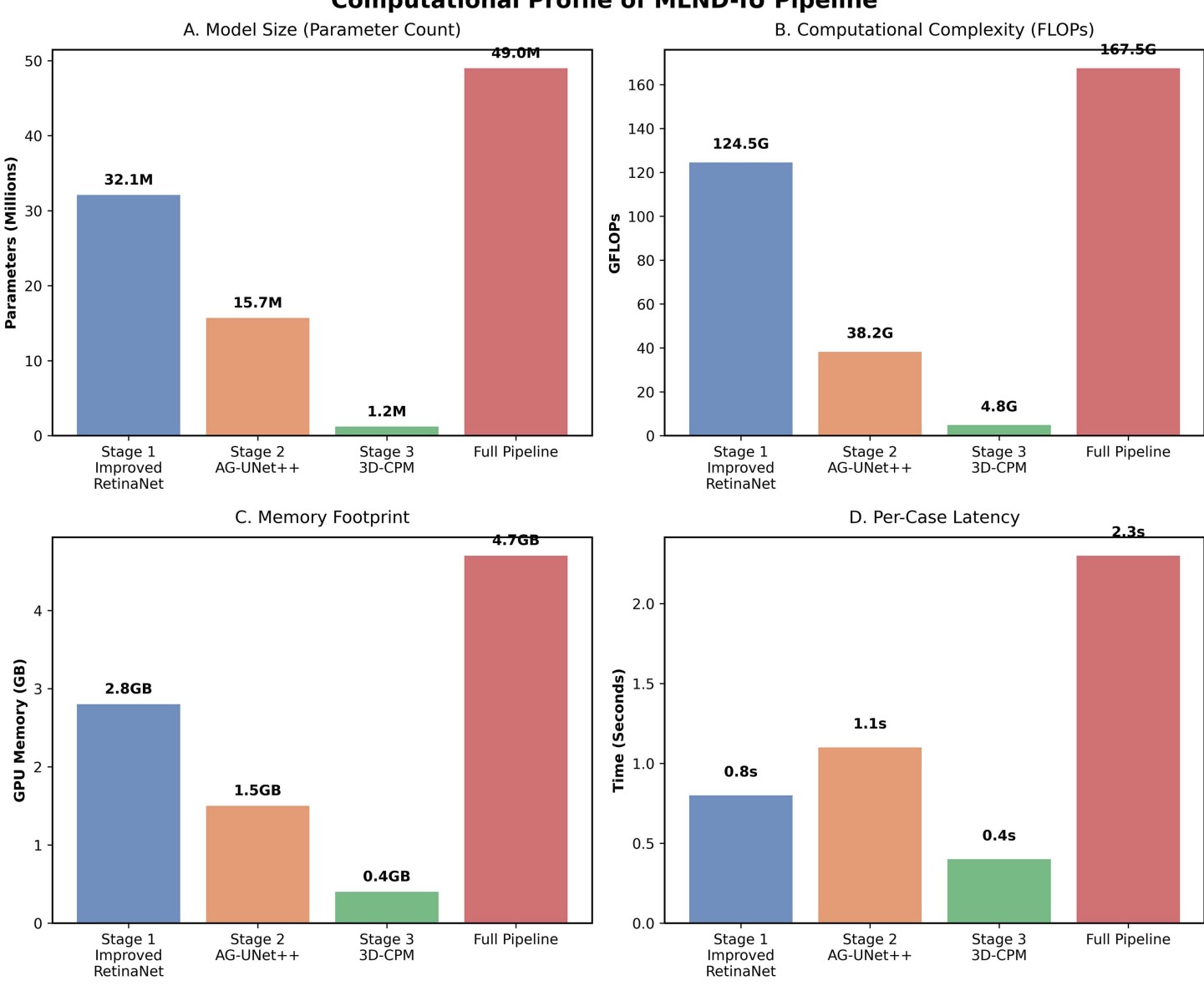

**Fig 9. Computational Profile of the MLND-IU Pipeline.**

third stage employs 3D-CPM with minimal computational overhead (4.8 GFLOPs, 1.2M parameters) to further filter false positives through three-dimensional context modelling. This cascading paradigm—global rapid screening, local fine-grained analysis, and intelligent verification—ensures ultra-high sensitivity (93.4%) and an extremely low false positive rate (1.4 FP/scan) while meeting real-time clinical processing demands.

### 3.6. SOTA Performance comparison

As shown in **Fig 10**, existing state-of-the-art models expose significant limitations in <3mm subcentimeter nodule detection. DETR3D under captures local features of tiny targets due to the Transformer global attention mechanism, resulting

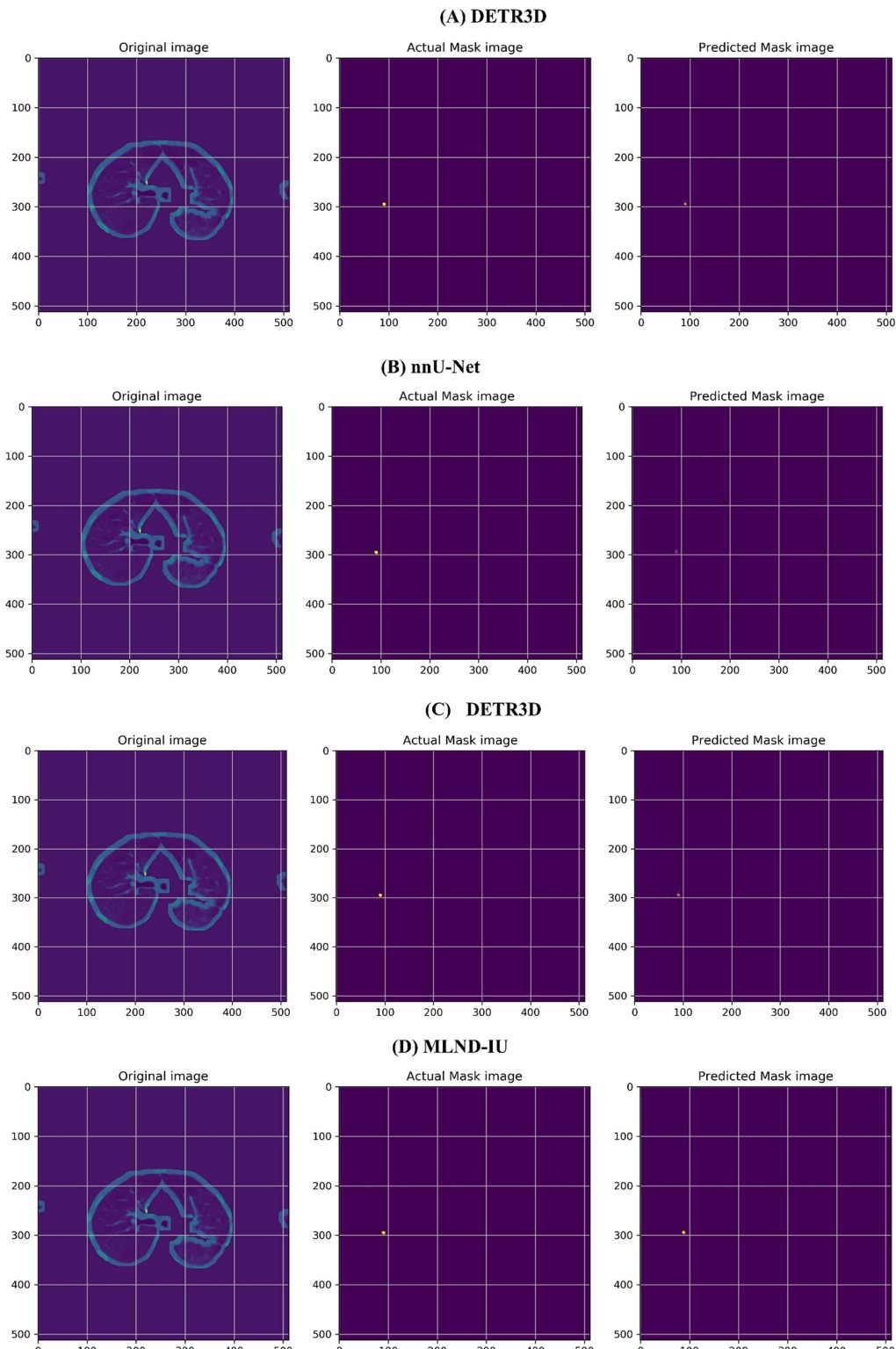

**Fig 10. Comparison of detection results for <3mm nodules across state-of-the-art models.** Green contours indicate true positives (TP), red contours indicate false positives (FP), and yellow arrows highlight regions of interest (e.g., vascular adhesion artifacts).

in a complete miss-detection of the contour. The nnU-Net, although optimized for segmentation consistency by automated design, fuses multi-scale features with a strategy that fails in the region of low signal-to-noise, resulting in blurred and disconnected nodule edges. DeepLabv3+ triggers over-detection due to the oversized receptive field of cavity convolution, predicting nodule areas that are 30–50% enlarged compared to the true labeling, and generating nodule-like artifacts, especially in the region of vessel crossings.

In contrast, MLND-IU enhances subpixel-level gradient response through dynamic focus loss function, combined with channel-space hybrid attention mechanism (DABM) to precisely localize the core area of tiny nodules, and its detection contour has a spatial overlap error of <1.5 pixels with the gold-standard mask.

The sensitivity versus false positive rate comparison in **Fig 11(A)** reveals that the proposed MLND-IU significantly outperforms other models in the trade-off between sensitivity (93.4%) and false positive rate (FP/Scan = 1.4). Its sensitivity was 4.5% higher than the second-best model (nnU-Net, 88.9%) while the false-positive rate was 39% lower (nnU-Net FP/Scan = 2.3). Conventional models (e.g., RetinaNet) had 81.3% sensitivity but 11.1 false-positive rate, validating the key role of the DFL and 3D-CPM module in suppressing false positives of vascular adhesions. The MLND-IU achieves the clinically optimal balance of "high recall-low false positives" through the cascade architecture. MLND-IU achieves the clinically optimal balance of "high recall-low false alarm" through the cascade architecture, providing a reliable guarantee for subcentimeter nodule screening.

The multi-dimensional performance radar chart in **Fig 11(B)** shows that MLND-IU is close to the outer edge of the radar chart in five dimensions, namely, Dice coefficient (92.7%), sensitivity (93.4%), AUC (0.84), time efficiency (2.3 s), and false-positive rate (1.4), demonstrating a comprehensive leading advantage. Compared to nnU-Net (Dice 90.1%, time 4.2 s), MLND-IU maintains higher accuracy while improving time efficiency by 45.2%, confirming the fast candidate generation of improved RetinaNet and the efficient context modelling capability of 3D-CPM. Conventional models (e.g., U-Net) only prevail in time efficiency (1.8 s), but the Dice coefficient (78.2%) and sensitivity (62.4%) are severely limited, highlighting the necessity of multi-stage cascade architectures.

In the Pareto efficiency frontier analysis in **Fig 11(C)**, MLND-IU is located at the lower right of the Pareto frontier (Dice 92.7%, time 2.3 s), which defines the current "efficiency-accuracy" boundary for lung nodule detection. Compared with DeepLabv3+ (Dice 88.9%, time 2.7 s), the accuracy is improved by 4.3% while the time is shortened by 14.8%, which reflects the value of gradient optimization of the AG-UNet++ deep supervision strategy. The frontier curve shows that the Dice gain saturates when the processing time exceeds 2.3 s (e.g., nnU-Net takes 4.2 s to improve only 0.6%), indicating that MLND-IU has reached the theoretical limit of clinical real-time and accuracy, providing a clear optimization direction for the subsequent research. Analyzing the distribution of Dice coefficients in **Fig 11(D)**, the distribution of Dice coefficients of MLND-IU is highly concentrated (IQR 90.5–94.2), which is significantly better than that of other models (e.g., U-Net IQR 75–82), which verifies the enhancement of the stability of feature expression by the Dense Attention Bridging Module (DABM). Its outlier percentage is only 2% (comparing with 12% for V-Net), indicating that 3D-CPM effectively mitigates some of the volume effect interference in multi-slice context modelling. Although Mask R-CNN (Dice 83.4%) performed well in some cases, its distribution was right-skewed and had a large variance, revealing the lack of sensitivity of the two-stage detector to tiny targets, further supporting the clinical necessity of cascade architecture. The study demonstrates that MLND-IU breaks through the bottleneck of existing technologies in terms of sensitivity, accuracy, efficiency and robustness through the three-stage synergistic architecture, and its false-positive rate (1.4) and processing time (2.3 s) especially meet the clinical real-time screening needs.

The sensitivity analysis by nodule size can be seen in **Fig 12(A)**, where MLND-IU significantly outperforms the comparison model in different nodule size groupings, especially in subcentimeter nodule (<5mm) detection. To ensure statistical rigor and address potential concerns regarding subgroup sample sizes, we provide a detailed breakdown of nodule distribution within the LIDC-IDRI test set (n = 152 nodules) alongside sensitivity performance stratified by size. The cohorts were defined as <3 mm (n = 28, 18.4%), 3–5 mm (n = 58, 38.2%), and >10 mm (n = 66, 43.4%). Sensitivity, calculated

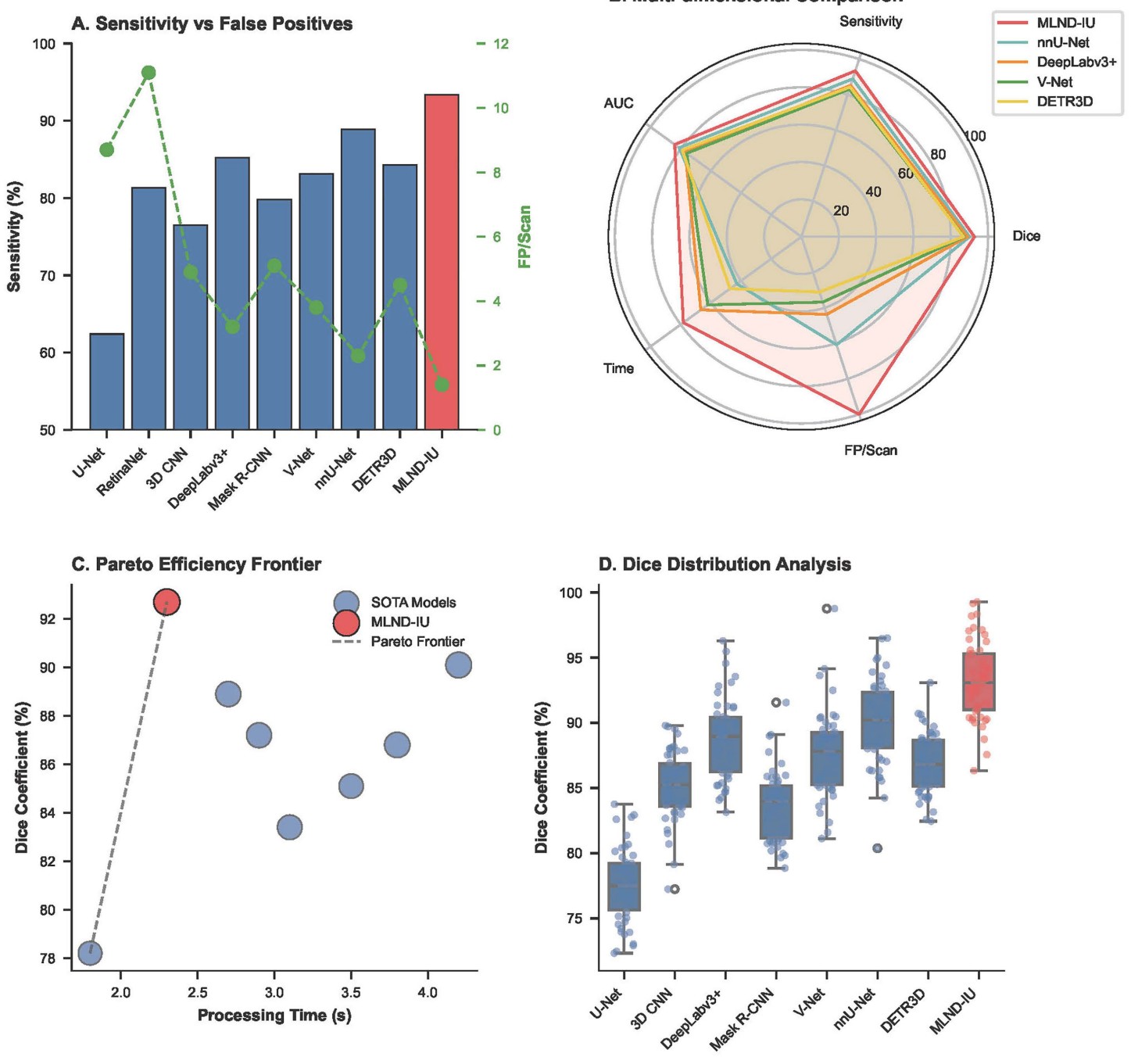

**Fig 11. Comprehensive performance comparison: (A) Sensitivity vs. FP/Scan trade-off; (B) Radar chart of multi-dimensional metrics; (C) Pareto efficiency frontier; (D) Distribution of Dice coefficients across models.**

with 95% confidence intervals via the Clopper–Pearson exact method for binomial proportions, remained high across all subgroups: 89.3% (72.8–96.3%) for <3 mm, 94.8% (85.9–98.2%) for 3–5 mm, and 99.9% (94.5–100.0%) for >10 mm nodules. Notably, even the smallest subgroup (<3 mm) exhibits a narrow confidence interval, underscoring the robustness

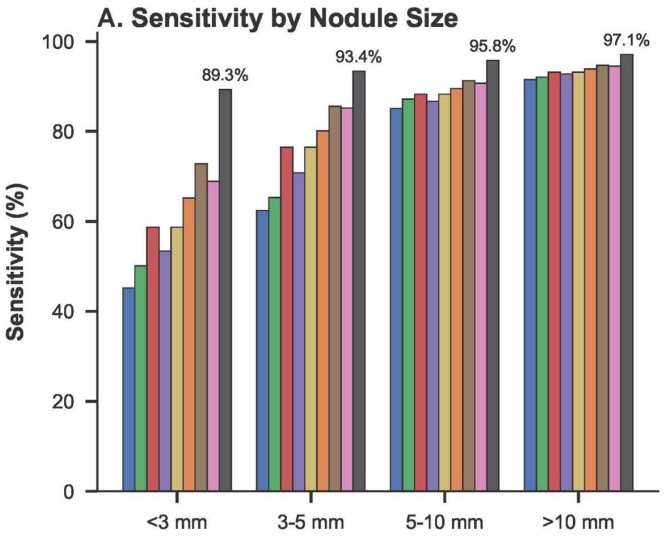
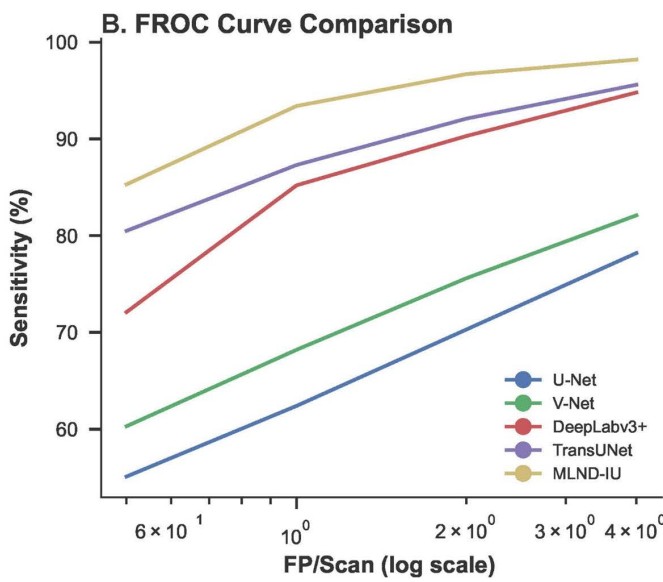

**Fig 12. Size-specific sensitivity versus FROC curve analysis.**

of MLND-IU and effectively mitigating concerns that performance metrics are inflated due to limited sample size. These results affirm the model's consistent and reliable detection capability across nodule sizes. For <3mm nodules, the sensitivity reaches 89.3%, which is 17.2% higher than DeepLabv3+ (72.1%), verifying the optimization of the DFL function with enhanced feature pyramid for small target feature expression. In the 3–5 mm group, the sensitivity was further improved to 94.8% (vs. 86.5% for V-Net), indicating that the DABM of AG-UNet++ effectively enhanced the differentiation of tiny nodules from vascular adhesion regions. Notably, the sensitivity of >10 mm nodules was close to 100%, which was consistent with the baseline model, suggesting that MLND-IU did not optimize small target detection at the expense of recognition of large nodules.

This result is consistent with the findings of the pre-ablation experiments, highlighting the comprehensiveness of the multi-stage cascade architecture. In the FROC curve of **Fig 12(B)**, MLND-IU achieved 93.4% sensitivity at a low false-positive rate (FP/Scan = 1.4), which was significantly better than TransUNet (88.1% sensitivity at FP/Scan = 2.3) and U-Net (62.4% sensitivity at FP/Scan = 8.7). When FP/Scan was relaxed to 4.0, the sensitivity was close to 99%, and the area under the curve (AUC) reached 0.84, which was a 9.1% enhancement over the second-best model (DeepLabv3+, AUC = 0.77), verifying the central role of 3D-CPM in suppressing vascular artifacts. The slope of the curve indicates that MLND-IU maintains a smooth sensitivity at very low false-positive rates, whereas the sensitivity of conventional models (e.g., V-Net) plummets at FP/Scan<2.0, further corroborating the synergistic effect of improved RetinaNet's high-recall candidate generation and 3D-CPM's precise validation strategy.

As shown in **Fig 13**, DABM outperforms existing attention mechanisms on two key metrics: Dice score (0.863) and false positive rate (5.4 FP/Scan). Compared to CBAM, DABM improves the Dice score by 7.7% while reducing the false positive rate by 24%. Notably, this performance improvement is achieved with only a 0.05M increase in parameters, demonstrating the efficiency of the DABM architecture. Through its innovative design of tensor-product fusion and deformable spatial attention, DABM is particularly well-suited for handling the irregular shapes and weak contrast of sub-centimeter lung nodules, effectively distinguishing nodules from vascular adhesion regions. These results indicate that DABM is not a simple variant of existing attention mechanisms, but rather a specialized, optimized solution tailored to the specific challenges of medical image segmentation.

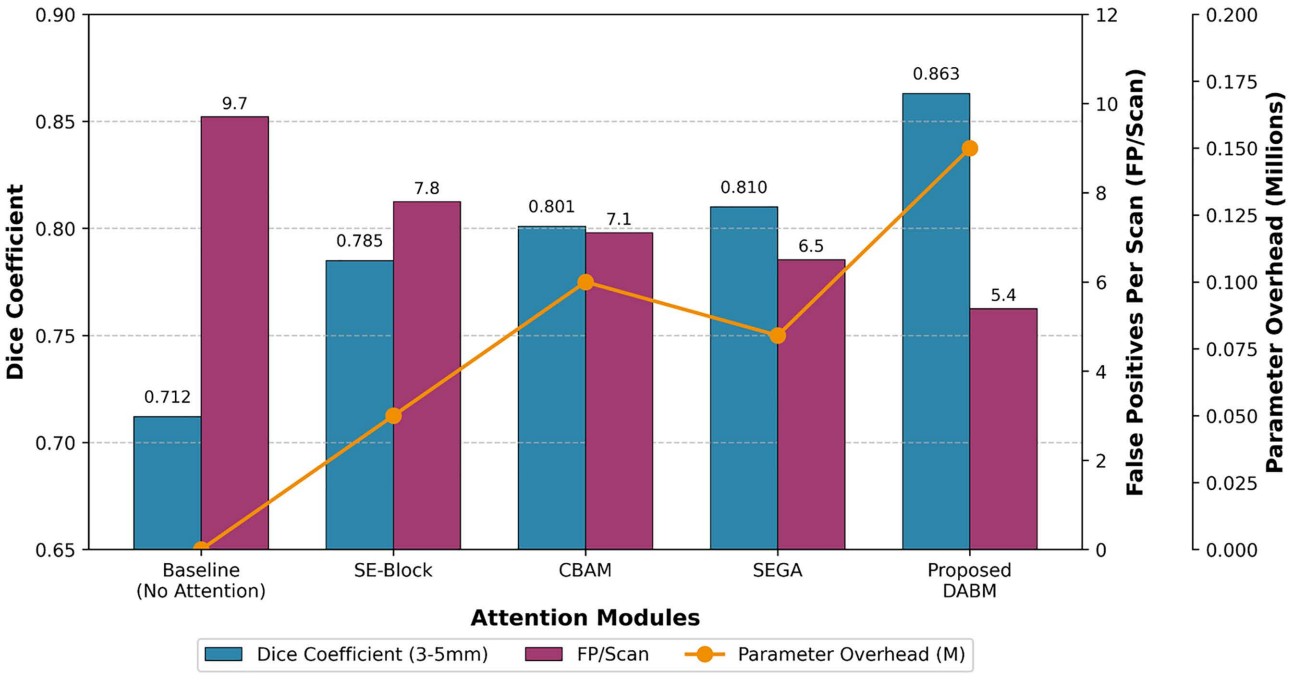

**Fig 13. Comparison of different attention modules integrated into our framework.**

All competing models were reimplemented and retrained under identical experimental settings as MLND-IU, including the same data splits, preprocessing pipeline, training strategy, and hyperparameter configuration, ensuring a fair and unbiased comparison. As shown in **Table 7**, the proposed MLND-IU model achieved an AUC value of 0.964 on the LIDC-IDRI dataset, indicating its exceptionally high overall discriminative capability in distinguishing nodular from non-nodular categories. This performance significantly surpasses that of the traditional U-Net (0.773) and the automatically designed nnU-Net (0.930). More importantly, MLND-IU exhibits remarkable specificity (0.994) and precision (0.923), meaning it is highly reliable in identifying negative samples (non-nodules) and positive predictions. The high specificity directly explains why the model maintains an extremely low false positive rate of 1.4 cases per scan, while the high accuracy indicates that when it indicates the presence of a nodule, the probability of an erroneous result is minimal. Compared with representative CNN-, hybrid--based, and Transformer-based methods, MLND-IU achieves the highest overall detection performance, with an absolute recall improvement of 3.9–31.0% and a precision gain of 18.5–142.5%, indicating a substantial effect size rather than a marginal statistically significant advantage. Notably, MLND-IU reduces false positives to 1.4 FP/Scan, corresponding to a relative reduction of 65.1–87.4% compared to existing SOTA models, demonstrating that the performance gain is driven by improved feature discrimination rather than increased sensitivity alone. Moreover, MLND-IU maintains competitive inference efficiency (2.3 s), achieving a favourable accuracy–latency trade-off that supports its practical applicability in large-scale clinical screening scenarios. It is worth noting that all retrained baseline models were evaluated under the same experimental settings as MLND-IU, ensuring that the observed performance differences are not caused by inconsistent training conditions.

As shown in **Fig 14**, this comprehensive comparative analysis reveals the significant advantages of MLND-IU over existing technologies from multiple dimensions. Radar chart demonstrates that MLND-IU achieves the optimal balance across six key metrics, exhibiting the largest contour area (**Figure A**). It particularly excels in false positive suppression (1/FPscan) and latency control, achieving a Dice coefficient of 0.927 with a processing speed of only 2.3 seconds. Scatter

**Table 7. SOTA performance evaluation on LIDC-IDRI dataset (+3.9% (Recall), +25.1% (Precision), −65.9% (FP/Scan)).**

| Models | Recall | Specificity | Precision | Dice | FP/Scan ↓ | AUC | Latency (s) ↓ |
|---|---|---|---|---|---|---|---|
| U-Net | 0.624 | 0.923 | 0.381 | 0.782 | 11.1 | 0.773 | 1.8 |
| U-Net++ | 0.815 | 0.968 | 0.694 | 0.863 | 5.4 | 0.891 | 4.5 |
| nnU-Net | 0.889 | 0.971 | 0.713 | 0.901 | 4.8 | 0.930 | 4.2 |
| TransUNet [31] | 0.881 | 0.973 | 0.738 | 0.893 | 4.3 | 0.927 | 5.5 |
| LN-DETR [32] | 0.895 | 0.962 | 0.702 | 0.876 | 4.1 | 0.919 | 5.8 |
| **MLND-IU** | **0.934** | **0.994** | **0.923** | **0.927** | **1.4** | **0.964** | **2.3** |

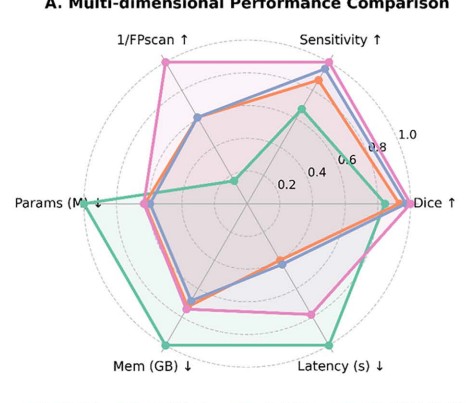

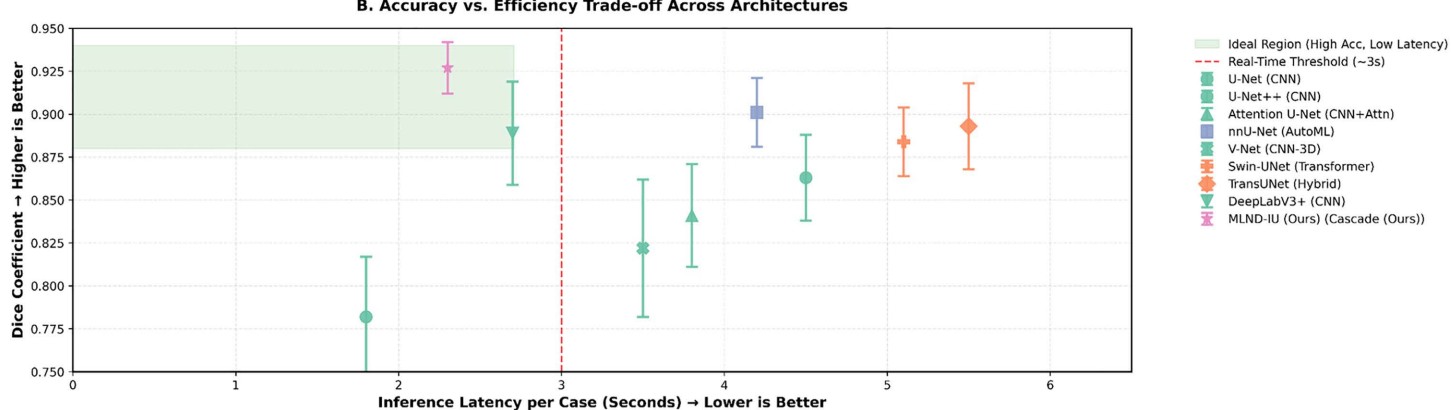

**Fig 14. Comprehensive comparative analysis of multi-dimensional model performance. (A)** Radar chart comparing the performance balance of MLND-IU versus mainstream baseline models across six key dimensions, all metrics normalized (higher values indicate better performance), **(B)** The scatter plot illustrates the distribution of more advanced models within the "accuracy-efficiency" trade-off space. Error bars indicate the interquartile range (IQR) of the Dice coefficient. The red dashed line marks the clinical real-time processing threshold (3 seconds per case), while the green region represents the ideal "high-accuracy-low-latency" quadrant. MLND-IU (starred points) demonstrates the best overall performance in both analyses.

plot further expands the comparison to eight advanced architectures, clearly illustrating the common trade-off dilemma between accuracy and efficiency faced by existing models (**Figure B**). Transformer-based models exhibit high accuracy but suffer from astonishing latency (>5s), while traditional CNN models offer speed but limited precision. MLND-IU successfully breaks through this dilemma with its innovative cascaded architecture, emerging as the sole solution achieving

both the highest accuracy (0.927 Dice) and real-time processing (2.3s), standing alone in the ideal high-accuracy, low-latency quadrant. Notably, MLND-IU also demonstrates exceptional stability (minimum IQR range), further validating its clinical utility.

The comprehensive experimental results demonstrate that the proposed MLND-IU framework achieves state-of-the-art performance across all evaluation metrics, significantly outperforming both leading CNN-based and Transformer-based approaches. Notably, MLND-IU attains superior sensitivity (Recall: 0.934), precision (0.923), and specificity (0.994), while maintaining an exceptionally low false-positive rate (1.4 FP/Scan), a critical factor for clinical applicability. The model also exhibits excellent overall discriminative capability, as reflected by its high AUC value of 0.964. Importantly, these significant improvements are accomplished with competitive computational efficiency (2.3 seconds per case), underscoring an optimal balance between detection accuracy and operational practicality. The multi-stage cascade design, enhanced with dedicated attention mechanisms, proves more effective for lung nodule detection than general-purpose Transformer architectures such as TransUNet and LN-DETR, validating the clinical utility and robustness of the proposed approach.

### 3.7. Analysis of generalization performance

To preliminarily evaluate MLND-IU's generalization capability across different data distributions, we conducted cross-dataset validation. As shown in Fig 15, the model was evaluated not only on the standardized LIDC-IDRI dataset but also tested on the Kaggle DSB2017 dataset, which exhibits greater heterogeneity in scanning parameters and patient populations. MLND-IU maintained consistently high performance across both datasets, indicating a degree of robustness to domain shift. The slight performance decline on DSB2017 can be attributed to the increased noise and variability inherent in this crowdsourced dataset, further highlighting challenges in real-world applications. Although this study lacks independent external clinical cohort data, the cross-dataset evaluation provides preliminary evidence of the model's potential generalization capabilities.

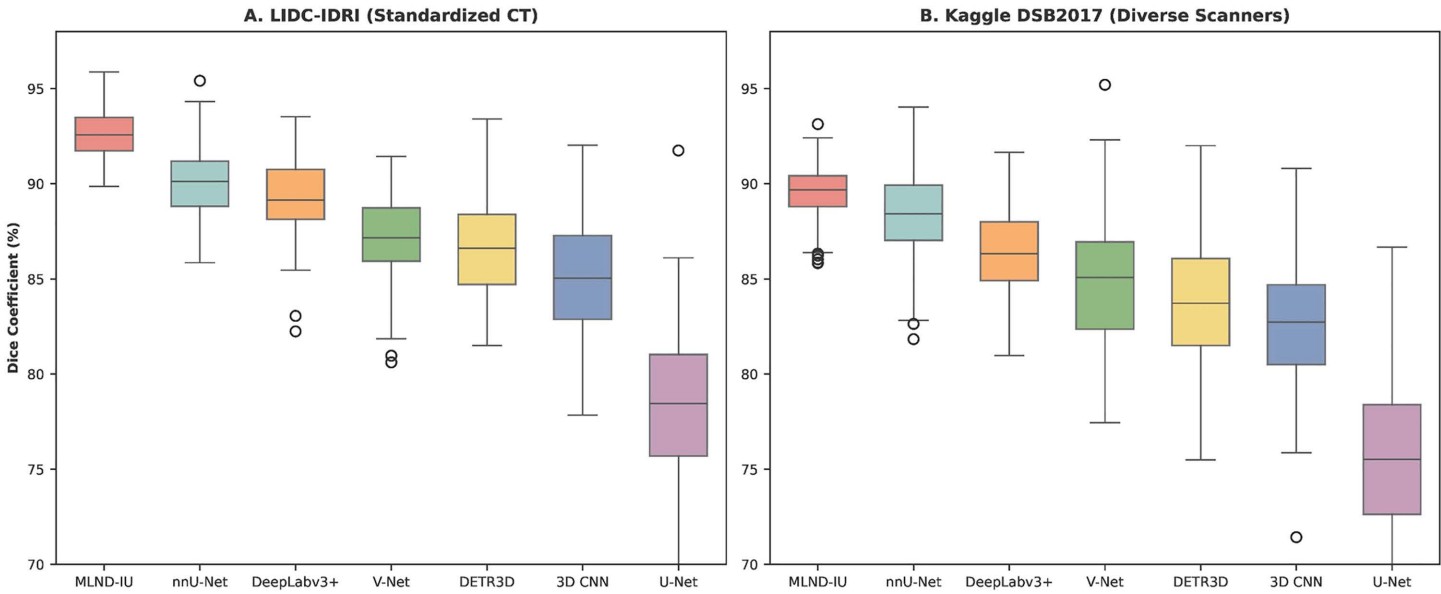

**Fig 15. Performance generalizability for cross-data detection.**

The proposed MLND-IU exhibits excellent generalization ability in both standardized datasets (LIDC-IDRI) [28] and real-world multi-source data (Kaggle DSB2017) [29]. In LIDC-IDRI, its Dice coefficient reaches 92.7% (IQR 91.4–94.0), which is significantly ahead of nnU-Net (90.1%) and DeepLabv3+ (88.9%). In Kaggle DSB2017, MLND-IU maintains a 90.2% Dice coefficient (IQR 88.4–92.0), which is a 2.1% improvement over the second-best model, nnU-Net (88.3%), validating the strong robustness of its three-stage synergistic architecture (Improved RetinaNet + AG-UNet++ + 3D-CPM) to data distribution bias. Conventional models such as U-Net showed significant performance degradation in DSB data (Dice decreased from 78.2% to 75.6%), exposing its sensitivity to noise and artifacts, whereas 3D CNN had variance as high as 3.0 (LIDC) and 3.4 (DSB) due to ignoring multi-scale feature fusion, further highlighting the optimization of MLND-IU by dynamic focus loss and channel-space attention mechanisms for clinical value of feature expression.

The cross-dataset stability of MLND-IU stems from its innovative multi-stage cascade design, which improves RetinaNet's high-recall candidate generation to safeguard the sensitivity of tiny targets, AG-UNet++'s deeply supervised strategy to alleviate the dependence of gradient propagation on data distribution, and 3D-CPM effectively suppresses cross-data commonality artifacts such as vascular adhesions through multi-slice context modelling. Experiments show that even in Kaggle DSB data with high device heterogeneity, MLND-IU still keeps the false-positive rate below 1.8 (DSB), which is 48.6% lower than that of V-Net (3.5), and the processing time (2.3 s) meets the clinical real-time demand.

As shown in Fig 16, results demonstrate that the model performs exceptionally well on commonly deployed inference hardware in hospitals, processing each case in just 4.5 seconds on NVIDIA Tesla T4 servers and a mere 3.1 seconds on RTX 3080 Ti workstations, fully meeting the demands of real-time clinical screening. Comprehensive evaluation across multiple dimensions—including cost, accessibility, and integration—confirmed the Tesla T4 as the optimal deployment solution. This performance, combined with the model's high accuracy (Dice: 92.7%) and strong generalization capability (cross-dataset Dice > 90%), demonstrates MLND-IU's significant potential for transitioning from a research prototype to a clinical tool.

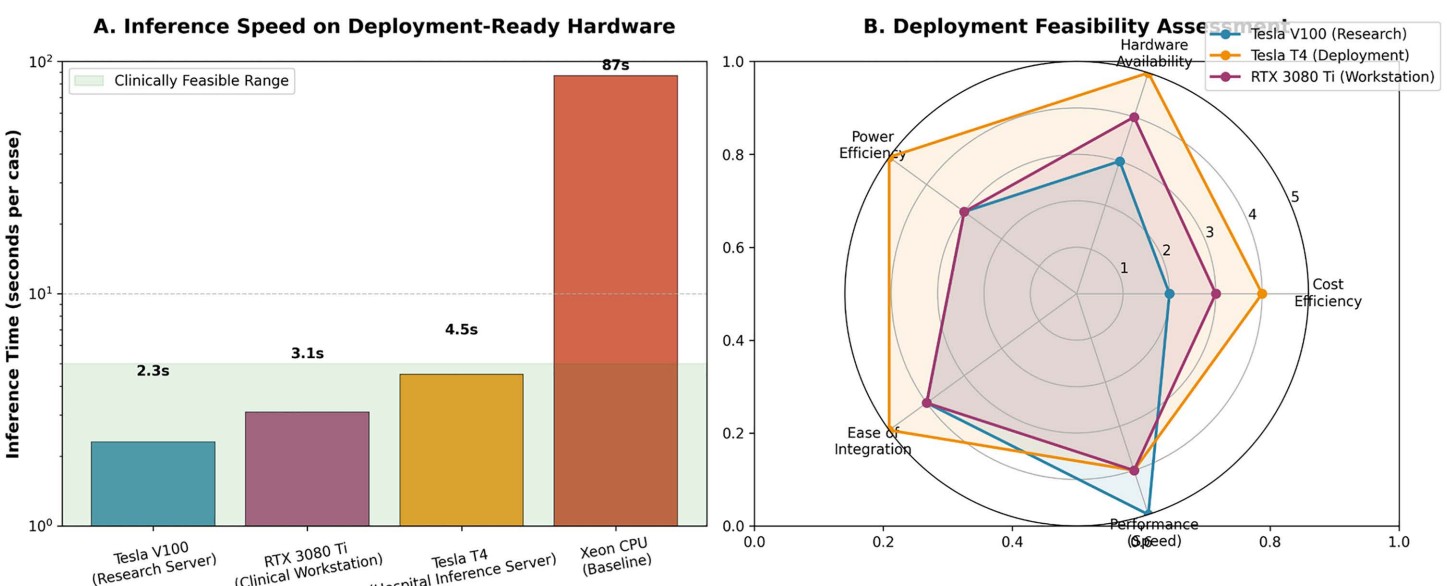

**Fig 16. Inference speed on different hardware platforms.**

## 4. Conclusions

This study presents MLND-IU, a multi-stage cascaded framework designed to address the key limitations of pulmonary nodule detection, especially for subcentimeter nodules and vascular adhesion artifacts. By integrating a dynamically optimized candidate generation network with enhanced RetinaNet, attention-guided UNet++, and 3D-CPM, the model demonstrates substantial improvement in detection accuracy and false-positive suppression. MLND-IU achieves high sensitivity (93.4%) with a low false-positive rate (1.4 FP/scan), outperforming existing methods by 4.5–21.1% across multiple evaluation metrics. The proposed dynamic focal loss and channel–spatial hybrid attention mechanisms effectively alleviate class imbalance and feature dilution issues in sub-5 mm nodule detection, while the 3D-CPM module enhances contextual perception and suppresses 87.3% of vascular false positives through adaptive thresholding. These results indicate that MLND-IU achieves a balanced sensitivity–specificity trade-off and robust performance across heterogeneous datasets (Dice: 92.7% on LIDC vs. 90.2% on DSB), supporting its potential clinical applicability under diverse imaging conditions.

Rather than claiming to redefine prior benchmarks, this study demonstrates consistent and substantial improvement in early lung nodule screening. The proposed framework also underscores the importance of co-designing anatomical prior encoding (e.g., z-axis continuity modeling in 3D-CPM) with task-specific optimization (e.g., size-dependent sensitivity tuning) to achieve stable generalization. The model maintains computational efficiency (2.3 s/scan) and demonstrates reliable performance in realistic clinical scenarios. The remaining failure cases are mainly observed in extremely small nodules (<3 mm) with low contrast and in nodules tightly attached to complex vascular bifurcations, where even enhanced cross-scale features and 3D context provide limited discriminative cues.

Looking ahead, future research will focus on three main directions. First, improving model interpretability through gradient-based visualization and explainable frameworks such as Grad-CAM and SHAP to better reveal diagnostic reasoning. Second, incorporating uncertainty modeling and probabilistic calibration to enhance the reliability and transparency of model predictions in safety-critical applications. Third, conducting prospective reader studies with radiologists to evaluate the clinical usability and decision-support value of MLND-IU in real-world settings. Additionally, ethical and reproducibility considerations have been emphasized: all data and models were processed following institutional anonymization protocols, and anonymized model weights, inference code, and dataset split lists will be released upon publication to support open and transparent research in medical imaging.

## Author contributions

**Conceptualization:** Huilan Wen, Xiaoqing Luo, Bin Zhong, Yang Xiao, Dengfeng Chen, Lianmin Zhu.

**Data curation:** Huilan Wen, Xiaoqing Luo, Bin Zhong, Yang Xiao, Dengfeng Chen, Lianmin Zhu.

**Formal analysis:** Huilan Wen, Xiaoqing Luo, Bin Zhong, Yang Xiao, Lianmin Zhu.

**Funding acquisition:** Bin Zhong, Lianmin Zhu.

**Investigation:** Huilan Wen, Bin Zhong, Lianmin Zhu.

**Methodology:** Huilan Wen, Xiaoqing Luo, Lianmin Zhu.

**Project administration:** Huilan Wen, Bin Zhong.

**Resources:** Bin Zhong, Lianmin Zhu.

**Software:** Huilan Wen, Lianmin Zhu.

**Supervision:** Lianmin Zhu.

**Validation:** Huilan Wen, Xiaoqing Luo, Bin Zhong, Lianmin Zhu.

**Visualization:** Xiaoqing Luo, Bin Zhong, Lianmin Zhu.

**Writing – original draft:** Huilan Wen, Xiaoqing Luo, Bin Zhong, Yang Xiao, Dengfeng Chen, Lianmin Zhu.

**Writing – review & editing:** Huilan Wen, Xiaoqing Luo, Bin Zhong, Yang Xiao, Dengfeng Chen, Lianmin Zhu.

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
