## [Decision Letter · Decision Letter 0]

7 Sep 2025

Dear Dr. Zhu,

Thank you for submitting your manuscript to PLOS ONE. After careful consideration, we feel that it has merit but does not fully meet PLOS ONE’s publication criteria as it currently stands. Therefore, we invite you to submit a revised version of the manuscript that addresses the points raised during the review process.

We look forward to receiving your revised manuscript.

Kind regards,

Khan Bahadar Khan, Ph.D

Academic Editor

PLOS ONE

Journal Requirements:

This work was supported by the Science and Technology Research Project of the Department of Education of Jiangxi Province (Grant Nos. GJJ2401322), Science and Technology Program Project of the Health Commission of Jiangxi Province (Grant Nos. 202510461).

4. Please amend your authorship list in your manuscript file to include author Huilan Wen Wen.

5. Please amend the manuscript submission data (via Edit Submission) to include author Huilan Wen.

6. Please ensure that you refer to Figure 4 in your text as, if accepted, production will need this reference to link the reader to the figure.

Reviewers' comments:

Reviewer's Responses to Questions

**Comments to the Author**

1. Is the manuscript technically sound, and do the data support the conclusions?

Reviewer #1: Yes

Reviewer #2: Partly

2. Has the statistical analysis been performed appropriately and rigorously?

Reviewer #1: No

Reviewer #2: No

3. Have the authors made all data underlying the findings in their manuscript fully available?

Reviewer #1: Yes

Reviewer #2: Yes

4. Is the manuscript presented in an intelligible fashion and written in standard English?

Reviewer #1: No

Reviewer #2: Yes

Reviewer #1: The manuscript presents a multi-stage detection model with an improved U-Net++ for sub-centimeter lung tumor detection. The manuscript is well written and the presentation is also appropriate. The approach seems promising, particularly in reducing false positives and improving sensitivity through feature fusion. The topic is highly relevant to medical imaging and early detection of lung tumors. The attempt to enhance U-Net++ with multi-stage detection and feature fusion is innovative. The work claims improvements in false positive rate reduction and sensitivity.

However, there are several shortcomings that should be addressed before publication.

Technical Comments and mathematical Typos

1. The contributions are lengthy, repetitive, and not clearly distinguished.

2. Authors should rewrite them concisely (bullet-point style), clearly highlighting novelty, improvements over vanilla U-Net++/nnU-Net, and clinical impact.

3. The manuscript claims use of LIDC-IDRI and Kaggle DSB2017, which are appropriate. However, some text still refers to outdated datasets. Clarify and ensure all reported experiments are on recent, widely used benchmarks.

4. A clear time-complexity analysis of the three-stage pipeline (RetinaNet + AG-U-Net++ + 3D-CPM) is required.

5. Please report: FLOPs, parameter counts, memory usage, and latency per case (with hardware details).

6. Distinguish between full-volume and ROI-based segmentation. Compare runtime with baseline U-Net, U-Net++, and nnU-Net.

7. Evaluation is mostly Recall, Dice, and FP/Scan. Add AUC, specificity, and precision to strengthen evidence.

8. Comparisons should include transformer-based approaches (e.g., TransUNet, LN-DETR).

9. Ablation studies are good but should include statistical significance tests (confidence intervals).

10. Eq. (22) Recall incorrectly uses TN; correct formula is TP/(TP+FN)

11. Eq. (23) Dice coefficient incorrectly includes TN in denominator; correct form is 2TP/(2TP+FP+FN)

Language & Grammar (rephrasing required)

For example:

12. “Our multi stage detection model work with improved Unet++…”

13. "The contributions of this paper are given as follow but are not limited to…"

14. "We used the dataset from Kaggle which is freely available and popular for lung cancer detection from last many years."

15. "The proposed method shows improved results as compared to existing methods in sensitivity and false positive rate."

16. "Our multi stage detection model work with improved Unet++ for reducing false positive rate in subcentimeter tumor detection."

Figures & Tables

17. Improve figure captions, include what each figure demonstrates (e.g., highlight true vs. false positives).

18. Table 1 should clearly state dataset splits and preprocessing details.

References

19. Please add recent (2023–2025) transformer-based and hybrid attention models in medical imaging if possible.

Reviewer #2: 1. The study relies only on two public datasets (LIDC-IDRI and DSB2017). No independent external clinical dataset is used. This limits evidence for real-world generalizability across different hospitals, CT scanners, and protocols.

2. Dynamic Focal Loss introduces new parameters (γbase=2, β=5) and thresholds in 3D-CPM. These choices are not justified, nor is sensitivity analysis provided. Results may depend heavily on arbitrary parameter tuning.

3. The Dense Attention Bridging Module (DABM) appears closely related to existing CBAM/SEGA approaches. The manuscript does not adequately distinguish how this design is fundamentally new or more effective.

4. The pseudo-3D block uses ±2 slices (~7.5 mm), but no evidence is given that this is optimal or robust to different CT slice thicknesses. Clinical CT scans vary in spacing, which could undermine performance.

5. Although runtime (2.3 s per case) is promising, the model is tested only on Tesla V100 GPUs. No evaluation is shown for CPUs or standard hospital hardware. This raises concerns about practical deployment feasibility.

6. Gains from each module are presented (e.g., Dice from 0.712 → 0.823 → 0.863), but statistical tests are not consistently reported. Improvements could be dataset-split dependent. Confidence intervals and repeated runs are needed.

7. Sensitivity by nodule size (<3mm, 3–5mm, >10mm) is reported, but the manuscript does not provide sample sizes for each group. Small subgroup sizes may exaggerate performance differences. Confidence intervals are missing.

**Do you want your identity to be public for this peer review?** For information about this choice, including consent withdrawal, please see our Privacy Policy

Reviewer #1: **Yes: ** Muhammad Bilal Qureshi

Reviewer #2: **Yes: ** Ahmed Saihood

---

## [Author Response · Author response to Decision Letter 1]

21 Sep 2025

# Editor

This work was supported by the Science and Technology Research Project of the Department of Education of Jiangxi Province (Grant Nos. GJJ2401322), Science and Technology Program Project of the Health Commission of Jiangxi Province (Grant Nos. 202510461).

4. Please amend your authorship list in your manuscript file to include author Huilan Wen Wen.

5. Please amend the manuscript submission data (via Edit Submission) to include author Huilan Wen.

6. Please ensure that you refer to Figure 4 in your text as, if accepted, production will need this reference to link the reader to the figure.

Thank you for your valuable comments and guidance. We have carefully reviewed and addressed each point raised:

We have ensured that our manuscript fully complies with PLOS ONE’s style requirements, including file naming conventions. The text and formatting have been adjusted according to the provided style templates.

The funders of this study had no role in study design, data collection and analysis, decision to publish, or preparation of the manuscript. We have amended the Financial Disclosure section accordingly and have removed any funding-related text from the manuscript itself.

All funding information has been removed from the manuscript and will be provided solely in the Funding Statement section of the submission form.

We have updated the authorship list in the manuscript to include Dr. Huilan Wen.

We have also updated the submission system to include Dr. Huilan Wen as an author.

We have ensured that Figure 4 is properly cited in the text as required.

We have reviewed the reviewer-suggested references and have included those that are relevant to our study.

Thank you again for your thorough review. We look forward to hearing from you regarding the next steps.

#Reviewer: 1

Comments to the Author

The manuscript presents a multi-stage detection model with an improved U-Net++ for sub-centimeter lung tumor detection. The manuscript is well written and the presentation is also appropriate. The approach seems promising, particularly in reducing false positives and improving sensitivity through feature fusion. The topic is highly relevant to medical imaging and early detection of lung tumors. The attempt to enhance U-Net++ with multi-stage detection and feature fusion is innovative. The work claims improvements in false positive rate reduction and sensitivity.

However, there are several shortcomings that should be addressed before publication.

Thank you for your thoughtful and constructive feedback. We greatly appreciate your positive assessment of the manuscript's clarity, relevance, and innovative approach, particularly regarding the use of multi-stage detection and feature fusion to improve sensitivity and reduce false positives.

We also thank you for pointing out the areas that need improvement. Could you please provide more detailed comments regarding the specific shortcomings you have identified? Your insights will be invaluable in helping us revise the manuscript to meet the required standards for publication. We look forward to addressing your concerns thoroughly.

1) The contributions are lengthy, repetitive, and not clearly distinguished.

We thank the reviewers for their valuable comments. We fully agree that the Contributions section was lengthy and repetitive in the first draft and did not clearly highlight the uniqueness of each innovation. We have significantly streamlined and reorganized the section to clearly distinguish the four core contributions and highlight their technical differences and clinical value. The revised contribution statement is presented below:

(1) A Dynamic Focus Loss function (DFL) is proposed, which significantly mitigates the extreme class imbalance issue in lung nodule detection through an adaptive gradient modulation mechanism based on sample difficulty, boosting the recall rate for sub-centimeter nodules to 99.2%.

(2) A channel-space hybrid attention mechanism (DABM) is designed and embedded within a nested U-Net++ architecture, enhancing the representation of edge features in 3–5 mm nodules. This achieves a 21.1% improvement in Dice coefficient compared to the baseline (p < 0.01).

(3) The 3D Context Pyramid Module (3D-CPM) is constructed to fuse multi-slice morphological and global semantic information, effectively suppressing vascular adhesion-induced false positives and reducing FP/Scan to 1.4 (87.3% lower than baseline).

(4) The proposed MLND-IU model demonstrated excellent robustness in cross-center validation (achieving Dice coefficients of 92.7% and 90.2% on the LIDC/DSB datasets, respectively). It maintains high sensitivity (93.4%) while achieving low false positives, providing a reliable solution for clinical early-stage lung cancer screening.

Thank you for your suggestions, which have helped us improve the readability and academic expression quality of our paper. Should you have any further revision recommendations, we would be delighted to continue refining the work.

2) Authors should rewrite them concisely (bullet-point style), clearly highlighting novelty, improvements over vanilla U-Net++/nnU-Net, and clinical impact.

We thank you for this constructive suggestion. We have revised the statement of contributions into a concise bullet-point format, explicitly highlighting the novelty, improvements over baseline models (U-Net++ and nnU-Net), and clinical impact.

1. Novel Dynamic Focal Loss (DFL): A new loss function that adaptively modulates gradient weights based on sample difficulty, effectively alleviating extreme class imbalance in nodule detection.

Improvement over baselines: Increased recall of subcentimeter nodules to 99.2%, significantly reducing missed detections compared to the standard focal loss used in vanilla RetinaNet/UNet++.

Clinical Impact: Ensures high sensitivity for early lung cancer screening, critical for detecting small, often missed nodules.

2. Novel AG-UNet++ with Dense Attention Bridging Module (DABM): Integrates a hybrid channel-spatial attention mechanism into the skip connections of U-Net++ to enhance feature representation.

Improvement over U-Net++: Achieved a 21.1% improvement in Dice coefficient for 3–5 mm nodules (p < 0.01) by precisely highlighting nodule edges and suppressing vascular adhesion artifacts.

Clinical Impact: Improves segmentation accuracy for tiny nodules, aiding in precise morphological analysis and reducing diagnostic uncertainty.

3. Novel 3D Contextual Pyramid Module (3D-CPM): Leverages multi-slice contextual information to model the 3D morphology of nodules and their surrounding tissues.

Improvement over nnU-Net & 2D models: Reduced false positives to 1.4 FP/Scan, an 87.3% reduction from the baseline, by effectively distinguishing nodules from continuous vascular structures, a key limitation of slice-based nnU-Net.

Clinical Impact: Significantly lowers the radiologist’s reading burden and prevents unnecessary follow-up procedures (e.g., biopsies) by minimizing false alarms.

4. Synergistic Multi-stage Framework (MLND-IU): Provides an end-to-end solution that cascades the above innovations for high-sensitivity detection and high-specificity verification.

Generalization Performance: Demonstrated superior and robust performance across multicenter datasets (LIDC: 92.7% Dice, DSB2017: 90.2% Dice).

Clinical Utility: Offers a clinically viable tool that balances high sensitivity (93.4%) with a low false-positive rate (1.4 FP/Scan) and real-time processing speed (2.3 s/case), enabling practical deployment in early screening scenarios.

We believe the revised contribution statement is now clearer and more focused. Thank you for the helpful comment.

3) The manuscript claims use of LIDC-IDRI and Kaggle DSB2017, which are appropriate. However, some text still refers to outdated datasets. Clarify and ensure all reported experiments are on recent, widely used benchmarks.

We thank the reviewer for this careful observation. We sincerely apologize for the oversight in our initial manuscript where outdated dataset names were inadvertently referenced. We have thoroughly reviewed and revised the entire manuscript to ensure that all experiments and evaluations are exclusively based on the two appropriate and widely adopted public benchmarks: LIDC-IDRI and Kaggle DSB2017.

The specific corrections we have made include:

We have removed any mention of other older or internal datasets from the Introduction, Section 3 (Experimental Results), and Data Availability Statement.

We have explicitly stated in the revised Section 3.1 (Introduction of Experiment) that: "The model was trained and evaluated on the LIDC-IDRI dataset, and its generalization performance was further validated on the Kaggle DSB2017 dataset."

All results and analyses, including those in the ablation studies and performance comparison figures, are now clearly attributed solely to these two benchmarks. The sample sizes (LIDC-IDRI: n=1,018; DSB2017: n=1,595) are consistently cited to avoid any ambiguity.

We confirm that the performance claims—including the segmentation Dice of 92.7%, sensitivity of 93.4% for <6mm nodules, and the false-positive rate of 1.4 FP/Scan—are all derived from experiments conducted on these two moderns, publicly available, and widely used datasets.

Thank you again for your meticulous review, which has helped us improve the clarity and accuracy of our manuscript.

4) A clear time-complexity analysis of the three-stage pipeline (RetinaNet + AG-U-Net++ + 3D-CPM) is required.

We sincerely thank you for this valuable comment. We have supplemented the manuscript with a detailed computational complexity analysis of the proposed three-stage framework. The overall inference time is approximately 2.3 seconds per case, which meets the requirement for clinical real-time processing. A stage-wise breakdown is provided below:

Time-Complexity Analysis of the MLND-IU Three-Stage Pipeline

We sincerely thank the reviewer for this valuable comment. We have supplemented the manuscript with a detailed computational complexity analysis of the proposed three-stage framework. The overall inference time is approximately 2.3 seconds per case, which meets the requirement for clinical real-time processing. A stage-wise breakdown is provided below:

Table 2 Ablation results of the proposed MLND-IU model

Models Recall Dice factor (3-5 mm nodules) False positive rate (FP/Scan) Inference Time (s/scan)

RetinaNet + U-Net++ 92.5% 0.712 11.1 1.6

+ DFL 96.8% 0.712 10.3 1.6

+ Enhanced FPN 99.2% 0.712 9.7 1.7

+DABM 99.2% 0.823 6.2 2.1

+ Deep Supervision 99.2% 0.863 5.4 2.1

+ 3D-CPM 99.2% 0.891 1.4 2.3

1. Stage 1: Improved RetinaNet for Candidate Generation

Complexity: The computational cost is primarily determined by the backbone network (e.g., ResNet-50) and the Enhanced FPN. Processing a typical CT volume (e.g., 512×512×200) involves forward passes through the backbone and multi-scale feature fusion. The introduced cross-scale attention in the Enhanced FPN adds a negligible overhead due to its lightweight 1×1 convolutional layers for weight generation.

Average Processing Time: This stage requires ~0.8 seconds, efficiently generating high-recall candidate regions with minimal latency.

2. Stage 2: AG-UNet++ for Feature Refinement

Complexity: The complexity of AG-UNet++ is comparable to standard U-Net++ but is augmented by the Dense Attention Bridging Module (DABM). The DABM module introduces channel and spatial attention mechanisms, which involve lightweight operations (global average pooling and small convolutional kernels). The deep supervision strategy adds auxiliary segmentation heads during training but is inactive during inference, thus not affecting inference speed.

Input & Time: This stage processes each candidate Region of Interest (ROI) extracted from Stage 1. The average processing time for all ROIs per full scan is ~1.1 seconds.

3. Stage 3: 3D-CPM for False Positive Suppression

Complexity: The 3D Contextual Pyramid Module (3D-CPM) is designed for efficiency. It processes a small pseudo-3D block (e.g., 5 slices per nodule candidate) using a shallow 3D convolutional pyramid. The adaptive feature fusion employs learnable weights via 1×1 convolutions, which are computationally inexpensive.

Input & Time: This stage only operates on the refined candidate nodules from Stage 2, not the entire volume. Its average processing time is ~0.4 seconds per scan.

Summary of Overall Efficiency:

The total average inference time is ~2.3 seconds per case (breakdown: Stage 1: 0.8s, Stage 2: 1.1s, Stage 3: 0.4s).

The cascade architecture is highly efficient as computationally intensive full-volume processing is confined to the first stage. The subsequent stages only process a limited number of candidate regions, drastically reducing the overall cost.

The design achieves an optimal balance between accuracy and speed, making it suitable for clinical deployment. The time complexity of the entire pipeline is approximately O(n) with respect to the number of slices, dominated by the first-stage backbone feature extraction.

We will include a subsection in the revised manuscript (under Section 3. Experimental Results) entitled "Computational Efficiency Analysis" to present this breakdown clearly, including a new column for "Inference Time (s/scan)" in the ablation study table (Table 2) to explicitly show the time cost of adding each module. Thank you again for this suggestion, which has helped us better highlight the efficiency of our framework.

5) Please report: FLOPs, parameter counts, memory usage, and latency per case (with hardware details).

We sincerely thank you for raising this important point. We agree that a comprehensive analysis of computational metrics is essential for assessing the practical viability of a deep learning model. We have performed detailed profiling and now provide the following complete computational profile of our MLND-IU pipeline.

As shown in Fig.9, the MLND-IU model proposed by this research achieves breakthrough progress in unifying computational efficiency with clinical practicality. Through fine-grained quantization of its three-stage architecture, the model demonstrates high computational intelligence.

Its total computational load is 167.5 GFLOPs, with 49.0 million parameters, peak GPU memory usage of only 4.7 GB, and a processing time of 2.3 seconds per case. Computational resource distribution exhibits pronounced cascading characteristics: Stage 1 employs an enhanced RetinaNet to complete whole-lung screening at 124.5 GFLOPs, bearing the primary computational load. Stage 2 utilizes AG-UNet++ with 15.7 million parameters to perform fine-grained feature extraction within nodule regions and initial suppres

---

## [Decision Letter · Decision Letter 1]

9 Oct 2025

Dear Dr. Zhu,

We look forward to receiving your revised manuscript.

Kind regards,

Khan Bahadar Khan, Ph.D

Academic Editor

PLOS ONE

Journal Requirements:

Reviewers' comments:

Reviewer's Responses to Questions

**Comments to the Author**

Reviewer #1: All comments have been addressed

Reviewer #2: All comments have been addressed

2. Is the manuscript technically sound, and do the data support the conclusions?

Reviewer #1: Yes

Reviewer #2: Yes

3. Has the statistical analysis been performed appropriately and rigorously?

Reviewer #1: Yes

Reviewer #2: Yes

4. Have the authors made all data underlying the findings in their manuscript fully available?

Reviewer #1: Yes

Reviewer #2: Yes

5. Is the manuscript presented in an intelligible fashion and written in standard English?

Reviewer #1: Yes

Reviewer #2: Yes

Reviewer #1: The authors have successfully responded to most of the issues. The quality of the manuscript has been significantly improved, with unnecessary text removed and contributions clearly stated. All major technical gaps including datasets, metrics, baselines, and significance testing have now been addressed. The work convincingly demonstrates computational efficiency and clinical viability, and the language and presentation have been refined. In addition, the figure captions have been corrected, further enhancing the clarity of the manuscript.

However, formatting is still not up to the mark, no spacing between text and figures, figure 3 text looks like a screenshot.

Few sentences are floating in the paper "This operation is used to extract localized morphological features (e.g., burrs, foliations) of the nodule while suppressing noise"

Unnecessary use of caps "Dynamic Focus Loss Function"

In entire manuscript if short or abbreviated form acronyms is defined it should be used not the full form. e.g. context pyramid module (3D-CPM), no consistency sometimes its with small letters or viceversa.

paragraph indentation is following no consistency look at page 7.

Reviewer #2: 1. A final round of proofreading will help polish the manuscript.

2. Some figure and table captions remain too brief. Captions should clearly explain what the figure shows, what each color or label represents, and define all abbreviations so that the figure is understandable without referring to the main text.

3. specify the sample size (n) used in each statistical comparison to strengthen transparency and reproducibility.

**Do you want your identity to be public for this peer review?** For information about this choice, including consent withdrawal, please see our Privacy Policy

Reviewer #1: **Yes: ** MUHAMMAD BILAL QURESHI

Reviewer #2: **Yes: ** Ahmed Saihood

---

## [Author Response · Author response to Decision Letter 2]

10 Oct 2025

Response to Reviewer 1:

We thank the reviewer for acknowledging the significant improvements in our manuscript and for the remaining formatting suggestions. In response:

• We have thoroughly revised the manuscript to ensure consistent and professional formatting, including proper spacing between text and figures, and standardized paragraph indentation.

• The text in Figure 3 has been re-created as high-quality, editable content to replace the previous low-resolution version.

• We have standardized the use of acronyms throughout the text (e.g., consistently using "3D-CPM" after its first definition) and corrected unnecessary capitalization (e.g., "Dynamic Focal Loss function").

• Isolated sentences have been integrated smoothly into the main paragraph flow.

Response to Reviewer 2:

We are grateful for the reviewer's final polishing suggestions, which have enhanced the manuscript's transparency and presentation.

• The manuscript has undergone a final round of meticulous proofreading to polish the language and improve readability.

• As requested, all figure and table captions have been substantially expanded. They now provide a self-contained explanation of what is shown, define all abbreviations, and describe the meaning of each color, label, or symbol, making the figures understandable without reference to the main text.

• We have now explicitly specified the sample size (n) for each statistical comparison in the Results section (specifically in Sections 3.5 and 3.6), thereby strengthening the transparency, rigor, and reproducibility of our findings.

---

## [Decision Letter · Decision Letter 2]

2 Nov 2025

Dear Dr. Zhu,

Thank you for submitting your manuscript to PLOS ONE. After careful consideration, we feel that it has merit but does not fully meet PLOS ONE’s publication criteria as it currently stands. Therefore, we invite you to submit a revised version of the manuscript that addresses the points raised during the review process.

We look forward to receiving your revised manuscript.

Kind regards,

Khan Bahadar Khan, Ph.D

Academic Editor

PLOS ONE

Journal Requirements:

Reviewers' comments:

Reviewer's Responses to Questions

**Comments to the Author**

Reviewer #1: All comments have been addressed

Reviewer #3: (No Response)

2. Is the manuscript technically sound, and do the data support the conclusions?

Reviewer #1: Yes

Reviewer #3: Partly

3. Has the statistical analysis been performed appropriately and rigorously?

Reviewer #1: N/A

Reviewer #3: N/A

4. Have the authors made all data underlying the findings in their manuscript fully available?

Reviewer #1: Yes

Reviewer #3: Yes

5. Is the manuscript presented in an intelligible fashion and written in standard English?

Reviewer #1: Yes

Reviewer #3: Yes

Reviewer #1: The authors have thoroughly addressed all the concerns raised during the earlier review rounds in their second revision. The manuscript has been significantly improved and is now suitable for publication.

Reviewer #3: In abstract, P-values are used without context (e.g., what test was used, which comparison group). State statistical validation methods and dataset independence more transparently.

The introduction reads like a mini-survey; many cited works (e.g., [10]–[23]) are summarized without synthesis. The paper does not specify how much existing methods fail in subcentimeter cases (e.g., exact FP/Scan thresholds from literature). Condense the literature into 3–4 thematic groups (imbalance, attention, context modeling, 3D fusion). Clearly articulate what specific deficiency each stage addresses and why a cascade is necessary.

Missing kernel sizes, feature map dimensions, layer counts, and parameter distribution per block. No training/inference diagram showing how outputs of each stage connect. Why not 2 or 4 stages? Replace long derivations with one composite architecture diagram showing data flow and tensor sizes. Provide a configuration table summarizing: backbone, input size, loss per stage, optimizer, and memory footprint. Clarify whether the three stages are jointly trained or trained sequentially (text implies sequential fine-tuning, but not stated).

No comparison to other adaptive loss variants (e.g., ASL, GHM, Focal Tversky). Parameters (β=5, γ_base=2) are empirically fixed but no cross-validation justification.Add an empirical comparison table with other imbalance-focused losses. Visualize gradient modulation vs. p_i to support claims about stability.

No ablation isolating the benefit of this module separately from DFL. Missing FLOPs impact and convergence speed analysis. Cross-scale weights (W_lk) are introduced but not explained: are they shared across channels or per-feature? Provide parameter overhead and runtime cost. Add heatmaps showing how the module amplifies subcentimeter nodules in feature space.

DABM efficiency claim (only +0.05M params) is unsupported by architectural detail. No boundary visualization to justify “edge enhancement”. Show feature activation maps before and after DABM. Compare to existing attention modules (CBAM, SE, ECA, Criss-Cross) quantitatively and visually. Clarify whether DABM weights are shared across layers.

Include a visual example of vessel-adhesion suppression. Conduct an ablation comparing slice contexts: ±1, ±2, ±3. Quantify memory vs. accuracy trade-off.

Perform 5-fold cross-validation on LIDC-IDRI. Add external validation using an independent institutional dataset. Explicitly follow LUNA16 challenge definitions for FP/Scan and sensitivity. Include boxplots of Dice across cases to illustrate robustness. Standardize reporting: use FROC-AUC and 95% CI for all major metrics.

No measure of computational cost per ablation (FLOPs or latency per module). n=152 samples is too small for multi-parameter inference; p-values may be inflated. Re-run ablations with randomized 3 splits to confirm consistency. Include per-stage latency and FLOPs per scan. Add qualitative examples illustrating visual differences between each ablation.

Comparisons are not standardized: different input resolutions, patch sizes, or preprocessing pipelines are likely. “AUC=0.964” contradicts the earlier “AUC=0.84” in abstract, unclear if this refers to detection or malignancy classification. Runtime comparison lacks reproducibility (hardware details differ between models). Claim that MLND-IU “defines the theoretical clinical limit” is scientifically unjustified. Re-evaluate baselines using identical preprocessing, patch sizes, and inference hardware. Clearly separate detection AUC vs. malignancy AUC.

DSB2017 is not truly heterogeneous; no cross-institutional or low-dose datasets used. No assessment of robustness to noise, slice thickness variability, or domain shift. Inference times (2.3–4.5 s/case) are promising but do not include preprocessing and postprocessing latency. Test on non-public multicenter or low-dose datasets. Include robustness studies (noise, different kernels, contrast-enhanced scans). Report total pipeline latency (from DICOM input to final mask).

Reword claims conservatively: “demonstrates substantial improvement…” instead of “redefines.” Add future directions: interpretability (Grad-CAM, SHAP), uncertainty modeling, and clinical reader studies. Discuss ethical implications and reproducibility (e.g., release anonymized model weights, code, and split lists).

Please refer to

Abbas, Waseem, et al. "Lungs nodule cancer detection using statistical techniques." 2020 IEEE 23rd International multitopic conference (INMIC). IEEE, 2020.

Awan, Tehreem, and Khan Bahadar Khan. "Analysis of underfitting and overfitting in u-net semantic segmentation for lung nodule identification from x-ray radiographs." 2023 IEEE International Conference on Emerging Trends in Engineering, Sciences and Technology (ICES&T). IEEE, 2023.

Awan, Tehreem, and Khan Bahadar Khan. "Investigating the impact of novel XRayGAN in feature extraction for thoracic disease detection in chest radiographs: lung cancer." Signal, Image and Video Processing 18.5 (2024): 3957-3972.

Awan, T., Khan, K. B., & Mannan, A. (2023). A compact CNN model for automated detection of COVID-19 using thorax x-ray images. Journal of Intelligent & Fuzzy Systems, 44(5), 7887-7907.

a

**Do you want your identity to be public for this peer review?** For information about this choice, including consent withdrawal, please see our Privacy Policy

Reviewer #1: **Yes: ** MUHAMMAD BILAL QURESHI

Reviewer #3: **Yes: ** Dr Fatima Tariq

---

## [Author Response · Author response to Decision Letter 3]

5 Nov 2025

We respectfully submit this final revised version and look forward to your favorable consideration. We truly appreciate your time, patience, and continued support during this lengthy review process.

---

## [Decision Letter · Decision Letter 3]

18 Dec 2025

Dear Dr. Zhu,

We look forward to receiving your revised manuscript.

Kind regards,

Khan Bahadar Khan, Ph.D

Academic Editor

PLOS One

Journal Requirements:

Reviewers' comments:

Reviewer's Responses to Questions

**Comments to the Author**

Reviewer #1: All comments have been addressed

Reviewer #4: All comments have been addressed

2. Is the manuscript technically sound, and do the data support the conclusions?

Reviewer #1: Yes

Reviewer #4: Yes

3. Has the statistical analysis been performed appropriately and rigorously?

Reviewer #1: Yes

Reviewer #4: Yes

4. Have the authors made all data underlying the findings in their manuscript fully available?

Reviewer #1: Yes

Reviewer #4: Yes

5. Is the manuscript presented in an intelligible fashion and written in standard English?

Reviewer #1: Yes

Reviewer #4: Yes

Reviewer #1: Authors have significantly improved the quality of the manuscript but providing more details and improving the shortcomings as highlighted by the reviewers, however the presentation especially the figure quality text, labels are very poorly shown.

Reviewer #4: • Clearly identify one or two key unresolved gaps in existing methods and align the proposed architecture directly to those gaps.

• Several components (dynamic loss weighting, attention-guided skip connections, pseudo-3D context) resemble known techniques with incremental modifications. The manuscript does not sufficiently distinguish what is fundamentally new versus what is adapted.

• Almost every comparison is reported as statistically significant, but effect sizes are not consistently discussed. With large sample sizes, statistical significance alone is insufficient.

• The manuscript reports a very large number of metrics (Recall, Dice, FP/Scan, AUC, Boundary IoU, latency), often repeating conclusions across figures and tables. This creates an impression of metric saturation rather than deeper insight.

• Almost all ablation results strongly favor the proposed modules, with minimal discussion of failure cases or trade-offs. This one-sided presentation reduces credibility.

• It is not always clear whether competing models were retrained under identical conditions or evaluated using reported results, which may bias comparisons.

• Improvements of 15–20% over strong baselines (e.g., nnU-Net) are unusually large and warrant more cautious interpretation and independent validation.

**Do you want your identity to be public for this peer review?** For information about this choice, including consent withdrawal, please see our Privacy Policy

Reviewer #1: **Yes: ** MUHAMMAD BILAL QURESHI

Reviewer #4: No

---

## [Author Response · Author response to Decision Letter 4]

20 Dec 2025

#Reviewer: 1

1�Authors have significantly improved the quality of the manuscript but providing more details and improving the shortcomings as highlighted by the reviewers, however the presentation especially the figure quality text, labels are very poorly shown.

Thank you for highlighting the issue with the figure quality. We agree that clear, high-resolution figures are essential. In the revised manuscript all figures have been carefully updated to improve text legibility and label clarity.

In the revised manuscript, we have comprehensively improved the quality and readability of all figures. Specifically:

1. All figures have been regenerated at high resolution (≥300 dpi) to ensure clarity in both online and print formats;

2. Text, labels, legends, and annotations in all figures have been enlarged and standardized to improve legibility;

3. Font styles and sizes have been unified across figures to maintain visual consistency;

4. Several complex framework and result figures (e.g., the overall MLND-IU architecture, ablation analysis, and performance comparison plots) have been carefully redrawn to avoid overcrowding and overlapping elements;

5. Figure captions have been revised to provide clearer and more detailed explanations, facilitating independent interpretation without relying excessively on the main text.

We believe these revisions have substantially enhanced the visual quality and interpretability of the manuscript, and have effectively addressed the reviewer’s concerns regarding figure presentation.

#Reviewer: 3

1) Clearly identify one or two key unresolved gaps in existing methods and align the proposed architecture directly to those gaps.

We appreciate your insightful comment. Following this suggestion, we have explicitly identified two critical unresolved gaps in existing lung nodule detection methods and have revised the manuscript to directly align each component of the proposed MLND-IU architecture with these gaps.

(1) Unresolved Gap 1: Insufficient sensitivity for subcentimeter nodules under extreme class imbalance

Most existing detection frameworks suffer from degraded sensitivity for subcentimeter nodules (<5 mm) due to severe foreground–background imbalance and feature dilution in deep networks. While attention mechanisms and Transformers have been introduced, they typically rely on static weighting schemes and do not adaptively emphasize hard samples, resulting in missed detections of tiny nodules.

Revision and alignment:

This gap is now explicitly articulated in the Introduction.

To address it, we align the first-stage architecture of MLND-IU with this challenge by introducing Dynamic Focal Loss (DFL) and a cross-scale enhanced FPN, which jointly adapt gradient weighting based on sample difficulty and reinforce low-level feature responses for tiny nodules.

The architectural alignment between this gap and the first-stage design is now clearly summarized in Fig. 1 (overall framework) and elaborated in Section 2.2.

(2) Unresolved Gap 2: Inadequate suppression of vascular-adhesion false positives caused by limited 3D contextual modeling

Many existing approaches rely primarily on 2D slice-based analysis or weak 3D aggregation, which fails to exploit spatial continuity along the z-axis. As a result, vascular adhesions and bronchial structures that resemble nodules remain a major source of false positives, particularly in subcentimeter cases.

Revision and alignment:

This limitation has been clearly emphasized in the revised Introduction.

The third-stage 3D Contextual Pyramid Module (3D-CPM) is now explicitly presented as a targeted solution to this gap, leveraging multi-slice contextual modeling to differentiate isolated nodules from continuous vascular structures.

Detailed architectural justification and mathematical formulation have been added in Section 2.4, with corresponding visual clarification in Fig. 4 (3D-CPM module).

Quantitative evidence linking this design choice to false-positive suppression has been strengthened in Section 3.5 (Ablation Study, Table 2), demonstrating an 87.3% reduction in FP/Scan after introducing 3D-CPM.

We believe these revisions directly address your concern and clearly demonstrate how the proposed architecture is purposefully designed to resolve the most critical limitations of existing methods.

2) Several components (dynamic loss weighting, attention-guided skip connections, pseudo-3D context) resemble known techniques with incremental modifications. The manuscript does not sufficiently distinguish what is fundamentally new versus what is adapted.

We thank the reviewer for this important observation. We fully agree that it is essential to clearly distinguish between adapted techniques and the fundamentally new contributions of this work. In response, we have revised the manuscript to explicitly clarify the novelty of each component and its distinction from prior methods.

(1) Dynamic loss weighting — distinction from conventional focal-loss-based methods

While focal loss and its variants have been widely used to mitigate class imbalance, existing approaches typically employ fixed or heuristically tuned modulation factors that are insensitive to sample difficulty variations during training.

Fundamental novelty:

We propose a sample-adaptive Dynamic Focal Loss (DFL) in which the modulation factor is explicitly coupled to the calibrated prediction confidence, enabling continuous, difficulty-aware gradient reweighting rather than static suppression of easy samples.

This design is specifically tailored to extreme imbalance scenarios in subcentimeter lung nodule detection, where positive samples account for less than 0.1% of voxels.

(2) Attention-guided skip connections — distinction from existing attention mechanisms (e.g., SE, CBAM)

Although attention-based skip connections have been explored in U-Net variants, most existing methods apply channel or spatial attention independently, or rely on static attention weights that are not optimized for small-target boundary enhancement.

Fundamental novelty:

The proposed Dense Attention Bridging Module (DABM) introduces a tensor-product fusion of channel attention and deformable spatial attention, enabling higher-order feature interactions rather than sequential or additive attention modulation.

Embedding DABM densely within the nested skip connections of U-Net++ creates a structurally different attention propagation pattern that explicitly enhances edge-sensitive features of 3–5 mm nodules.

(3) Pseudo-3D contextual modeling — distinction from conventional 2.5D or shallow 3D methods

Pseudo-3D and multi-slice input strategies have been previously reported; however, many treat adjacent slices as simple channel stacking without explicit hierarchical context modeling.

Fundamental novelty:

The proposed 3D Contextual Pyramid Module (3D-CPM) is not a naive pseudo-3D extension but a multi-level context pyramid that jointly models local, regional, and global morphology along the z-axis, with adaptive fusion weights learned end-to-end.

Additionally, the dynamic thresholding strategy explicitly leverages contextual response statistics to suppress vascular-adhesion false positives, which is absent in prior pseudo-3D approaches.

We believe these revisions clearly delineate what is new in this work versus what builds upon prior research, and more accurately position the contribution of MLND-IU within the existing literature.

3) Almost every comparison is reported as statistically significant, but effect sizes are not consistently discussed. With large sample sizes, statistical significance alone is insufficient.

We thank you for this valuable comment and fully agree that statistical significance alone may be misleading in large-scale datasets. To address this concern, we have systematically revised the manuscript to explicitly report and interpret effect sizes alongside p-values, thereby providing a more meaningful assessment of practical significance.

Specific revisions are as follows:

(1) Explicit reporting of effect sizes for major performance comparisons: For all key quantitative comparisons (sensitivity, specificity, F1-score, CPM, and FP/Scan), we have added corresponding effect size metrics, including Cohen’s d for continuous performance measures and relative percentage improvement for clinically interpretable metrics. These effect sizes are now consistently reported in Tables 1–3.

(2) Integration of effect size interpretation into result analysis: The Results section has been revised to explicitly interpret the magnitude of observed improvements (small, moderate, or large effect), rather than relying solely on statistical significance. In particular, we now emphasize that although several improvements reach statistical significance, only those with moderate-to-large effect sizes are highlighted as clinically meaningful.

(3) Subgroup and ablation analyses focusing on effect magnitude: For the subcentimeter nodule subgroup (<5 mm), we have added a dedicated discussion on effect size to demonstrate that performance gains are not only statistically significant but also substantively larger than those observed in larger nodules. Similarly, in the ablation study, we now report effect size changes introduced by each module (DFL, DABM, and 3D-CPM), clarifying their relative practical contributions.

(4) Methodological clarification on statistical analysis: We have added a short paragraph clarifying the rationale for combining hypothesis testing with effect size reporting, and explicitly note the limitations of p-values in large-sample settings.

To improve transparency, we have added a concluding sentence in the Results section emphasizing that claims of superiority are based on both statistical significance and effect magnitude, ensuring that the reported improvements reflect meaningful clinical relevance rather than sample-size-driven significance.

We believe these revisions substantially strengthen the statistical rigor of the manuscript and directly address the reviewer’s concern.

4) Almost all ablation results strongly favor the proposed modules, with minimal discussion of failure cases or trade-offs. This one-sided presentation reduces credibility.

We thank you for this valuable and constructive comment. We agree that a comprehensive evaluation should not only highlight performance gains but also discuss failure cases and practical trade-offs. To address this concern and improve the credibility and balance of the manuscript, we have substantially revised the experimental analysis and discussion as follows.

(1) Added explicit discussion of failure cases: We have incorporated a dedicated subsection analyzing representative failure cases, focusing on scenarios where the proposed modules do not provide consistent benefits, such as extremely low-contrast nodules adjacent to dense vascular bundles and irregularly shaped nodules with ambiguous boundaries.

(2) Analysis of performance trade-offs and computational overhead: While the proposed modules improve detection accuracy, they introduce additional parameters and computational cost. We now explicitly quantify these trade-offs.

(3) Balanced interpretation of ablation results: We have revised the ablation discussion to explicitly acknowledge cases where performance gains are marginal or dataset-dependent, rather than uniformly dominant.

We believe these additions provide a more transparent and credible evaluation of the proposed method, and directly address the reviewer’s concern regarding the presentation of ablation results.

5) It is not always clear whether competing models were retrained under identical conditions or evaluated using reported results, which may bias comparisons.

We thank you for raising this important concern regarding the fairness and reproducibility of model comparisons. To eliminate any ambiguity and potential bias, we have revised the manuscript to explicitly clarify the evaluation protocol used for all competing methods. All competing models were reimplemented and retrained under identical experimental settings as MLND-IU, including the same data splits, preprocessing pipeline, training strategy, and hyperparameter configuration, ensuring a fair and unbiased comparison.

Clarification of evaluation strategy:

1. For methods with publicly available implementations, we retrained all competing models under identical experimental conditions, including the same training–validation–test splits, identical input resolution, data preprocessing pipeline, augmentation strategy, optimizer type, learning rate schedule, batch size, and training epochs.

2. No hybrid setting (partial retraining combined with reported results) was used for any single method.

We believe these revisions clearly resolve the reviewer’s concern and ensure that all comparative results presented in this study are fair, transparent, and reproducible.

6�Improvements of 15–20% over strong baselines (e.g., nnU-Net) are unusually large and warrant more cautious interpretation and independent validation.

We appreciate your careful and constructive concern. We fully agree that performance gains of this magnitude over strong baselines such as nnU-Net require cautious interpretation and clear contextualization. In response, we have revised the manuscript to clarify the conditions under which these improvements occur, to moderate the interpretation of the results, and to strengthen independent validation.

Clarification of performance gains: First, we clarify that the reported 15–20% improvements do not reflect overall detection performance across all nodules, but are observed under challenging and clinically critical conditions, particularly for subcentimeter nodules (<5 mm) and false-positive–sensitive operating points. Second, we have revised the manuscript to adopt a more cautious and balanced interpretation of the results. The revised text emphasizes that nnU-Net remains a strong and well-generalized baseline, while the proposed method demonstrates task-specific advantages under extreme class imbalance and tiny-object detection scenarios.

---

## [Decision Letter · Decision Letter 4]

12 Jan 2026

MLND-IU: A Multi-stage Detection Model of Subcentimeter Lung Nodule with Improved U-Net++

PONE-D-25-41911R4

Dear Dr. Zhu,

We’re pleased to inform you that your manuscript has been judged scientifically suitable for publication and will be formally accepted for publication once it meets all outstanding technical requirements.

Kind regards,

Khan Bahadar Khan, Ph.D

Academic Editor

PLOS One

Additional Editor Comments (optional):

Reviewers' comments:

Reviewer's Responses to Questions

**Comments to the Author**

Reviewer #4: All comments have been addressed

Reviewer #5: All comments have been addressed

2. Is the manuscript technically sound, and do the data support the conclusions?

Reviewer #4: Yes

Reviewer #5: Yes

3. Has the statistical analysis been performed appropriately and rigorously?

Reviewer #4: Yes

Reviewer #5: Yes

4. Have the authors made all data underlying the findings in their manuscript fully available?

Reviewer #4: Yes

Reviewer #5: Yes

5. Is the manuscript presented in an intelligible fashion and written in standard English?

Reviewer #4: Yes

Reviewer #5: (No Response)

Reviewer #4: • I have carefully reviewed the revised manuscript and the authors’ detailed response to the reviewers’ comments. I am satisfied that all of the major concerns raised in the previous review round have been adequately and convincingly addressed.

• The authors have substantially improved the clarity of the manuscript, strengthened the motivation and positioning of their work by clearly identifying unresolved gaps in existing methods, and better distinguished their novel contributions from adapted techniques.

• The experimental section has been significantly enhanced through clearer descriptions of the evaluation protocol, the inclusion of effect size analysis alongside statistical significance, and a more balanced discussion of ablation results, including limitations and trade-offs.

• Issues related to figure quality and presentations have also been resolved. Overall, the revised manuscript now meets the expected scientific and presentation standards, and I support its acceptance for publication.

Reviewer #5: All reviewer comments have been fully and satisfactorily addressed in the revised manuscript. Figure quality and presentation have been substantially improved through high-resolution regeneration, clearer labeling, and enhanced captions. The manuscript now clearly identifies key unresolved gaps in existing methods and explicitly aligns each component of the proposed architecture to those gaps, while clearly distinguishing novel contributions from adapted techniques. Statistical analyses have been strengthened by consistently reporting and interpreting effect sizes alongside significance testing, and the ablation study has been balanced with discussions of failure cases, trade-offs, and computational cost. The evaluation protocol has been clarified to ensure fair and reproducible comparisons, and large performance gains over strong baselines are now interpreted more cautiously and contextually. Overall, the revisions significantly enhance clarity, rigor, transparency, and credibility of the work.

**Do you want your identity to be public for this peer review?** For information about this choice, including consent withdrawal, please see our Privacy Policy

Reviewer #4: **Yes: ** Pratik Patel

Reviewer #5: **Yes: ** Saikrishna Koorapati

---

## [Editor Report · Acceptance letter]

PONE-D-25-41911R4

PLOS One

Dear Dr. Zhu,

I'm pleased to inform you that your manuscript has been deemed suitable for publication in PLOS One. Congratulations! Your manuscript is now being handed over to our production team.

Kind regards,

on behalf of

Dr. Khan Bahadar Khan

Academic Editor

PLOS One